# The Value of Hydroclimatic Teleconnections for Snow-based Seasonal Streamflow Forecasting in Central Asia

Atabek Umirbekov[1,4], Mayra Daniela Peña-Guerrero[1,4], Iulii Didovets[2], Heiko Apel[3], Abror Gafurov[3] and Daniel Müller[1, 4, 5]

[1] Leibniz Institute of Agricultural Development in Transition Economies (IAMO), Theodor-Lieser-Str. 2, 06120 Halle (Saale), Germany
[2] Potsdam Institute for Climate Impact Research (PIK), Telegrafenberg A 31, 14473 Potsdam, Germany
[3] GFZ Helmholtz Centre for Geoscience, Telegrafenberg, 14473 Potsdam, Germany
[4] Geography Department, Humboldt-Universität zu Berlin, Unter den Linden 6, 10099 Berlin, Germany
[5] Integrative Research Institute on Transformations of Human-Environment Systems (IRI THESys), Humboldt Universität-zu-Berlin, Unter den Linden 6, 10099 Berlin, Germany

*Correspondence to*: Atabek Umirbekov (umirbekov@iamo.de)

**Abstract.** Due to the long memory of snow processes, statistically based seasonal streamflow prediction models in snow-dominated catchments can successfully leverage, but also typically rely on, snowpack estimates. Using mountainous catchments in Central Asia as a case study, we demonstrate how seasonal hydrological forecasts benefit from incorporating large-scale climate oscillations (COs). First, we examine the teleconnections between the major COs and peak precipitation season in eight catchments across the Pamir and Tian-Shan mountains from February to June. We then employ a machine learning framework that incorporates snow water equivalent (SWE) and dominant COs indices as predictors for mean discharge from April to September. Our workflow leverages an ensemble technique that uses multiple SWE estimates from near-time global data sources and diverse types of explainable machine-learning models. We find that the winter states of the El Niño-Southern Oscillation and the North Atlantic Oscillation enhance SWE-based forecasts of seasonal discharge in the study catchments. We identify three instances in which the inclusion of COs as additional predictors could be instrumental for snowpack-based seasonal streamflow forecasting: 1) when forecasts are issued at extended lead times and accumulated SWE is not yet representative of seasonal terrestrial water storage; 2) when climate variability during the forecasted season plays a larger role in shaping seasonal discharge; and 3) SWE estimates for a catchment are subject to larger uncertainty. Our approach provides a useful way to reduce uncertainties in seasonal discharge predictions in data-scarce snowmelt-dominated catchments.

## 1. Introduction

Snowmelt-driven streamflow is a vital source of water supply for downstream regions around the globe, sustaining ecosystems, agriculture, hydropower, and numerous human activities (Immerzeel et al., 2020; Viviroli et al., 2007). Around two billion people live in snow-sensitive basins (Mankin et al., 2015). Projections suggest that around a quarter of the world`s lowland population will be critically dependent on snow- and glacier-melt runoff from mountains by the middle of the century (Viviroli et al., 2020). Accurate water supply forecasts are essential for the sustainability and resilience of water-dependent human and ecological systems in these regions.

Seasonal streamflow forecasts are usually generated using either process-based or data-driven approaches. Process-based, dynamical forecasts encompass a hydrological or land-surface model to estimate initial hydrologic conditions, typically with the assimilation of observational data, followed by climate forecasts to project future conditions (Troin et al., 2021). One major advantage of process-based approaches is the continuous production of future streamflow states (Modi et al., 2022). On the other hand, a limitation of dynamical forecasts is their dependence on spatially distributed meteorological variables obtained from numerical climate models, which are prone to uncertainties. In addition, process-based approaches typically have higher computational demands. Meanwhile, data-driven approaches rely on the empirical relationship between one or multiple

predictor variables and seasonal streamflow. In this respect, data-driven hydrological forecasts in snow-dominated catchments offer advantages in terms of lower computational complexity and reliance on initial hydrological conditions. Both process-based and data-driven models for water supply forecasting can ingest seasonal to subseasonal climate forecasts as input. Process-based models offer an advantage of explicitly representing physical processes, improving their credibility and interpretability. In contrast, data-driven models do not rely on predefined physical assumptions, allowing for greater flexibility in capturing relationships without the need for explicit process representation in a computational framework.

Because accumulated snowpack is a main source of predictability of river streamflow in snowmelt-dominated basins (Pechlivanidis et al., 2020), statistical forecasts of seasonal streamflow often rely on accumulated snowpack, with use of additional predictors that contribute to estimation of initial hydrological conditions. North America has the longest history of systematically developing seasonal streamflow forecasts, also known in this domain as *water supply forecasts*, using empirical relationships between accumulated snow and spring-summer runoff. While snowpack explains most seasonal streamflow variability in western US basins, climate variability after the forecast issuance date has been identified as the main source of forecast error (Church, 1935, as cited in Pagano & Garen, 2010; Schaake & Peck, 1985). Early attempts to use climate oscillation indices in water supply forecasts began in the 1970s, integrating them into operational water supply forecasting at some agencies at the beginning of the 2000s, though widespread adoption did not follow immediately (Pagano and Garen, 2010).

Previous research has shown that integrating climate indices generally improves seasonal streamflow forecasts. Studies from the U.S. suggest that the improvement may be more evident in long-lead forecasts, as climate indices tend to account for future climatic conditions after the forecast issuance dates (Grantz, Rajagopalan, Clark, & Zagona, 2005; Hamid & Matthew, 2010; Kalra, Ahmad, & Nayak, 2013; Kennedy, Garen, & Koch, 2009; Regonda, Rajagopalan, Clark, & Zagona, 2006). Similarly, evidence from parts of High Mountain Asia suggests that climate indices may be better predictors than snowpack for streamflow forecasting in snowmelt-dominated catchments at the beginning of winter (Charles et al., 2018; Umar et al., 2023). While confirming that snowpack is one of the main predictors, a multi-model ensemble study on seasonal streamflow forecasting in the Andes also highlights the utility of large-scale ocean-atmospheric factors as additional predictors (Mendoza et al., 2014). However, the improvement from combining climate indices into snowpack-based streamflow forecasts depends on the strength of the teleconnections with large-scale climate oscillations (Mendoza et al., 2017; Opitz-Stapleton et al., 2007). In turn, relationships between large-scale climate oscillations and hydrometeorological variability maybe nonlinear and non-monotonic, making them challenging to capture with linear approaches (Fleming and Dahlke, 2014).

From a methodological perspective, data-driven seasonal streamflow forecasting has undergone two major transformations over the past decades. Historically dominated by the use of linear regression and its extensions, such as Principal Component Regression, the field increasingly adopts machine learning (ML) techniques. ML-based data-driven approaches generally excel at leveraging diverse datasets, capturing non-linear relationships, and achieving higher predictive accuracy in seasonal streamflow forecasting (Sean W Fleming, Rittger, Oaida Taglialatela, & Graczyk, 2024; Kalra et al., 2013; Korsic et al., 2023). Another notable trend is the growing use of ensemble approaches because they generally offer higher prediction accuracy and allow quantification of prediction uncertainty compared to single-model methods (Murray, 2018; Zounemat-Kermani et al., 2021). A notable example that integrates these two trends for forecasting in snowmelt-dominated catchments is the multi-model machine-learning metasystem ("M4") in the western US, which uses an ensemble approach with multiple ML-based forecast models and pool outputs to generate a consensus prediction (Fleming et al., 2021). Furthermore, Najafi and Moradkhani (2016) explored techniques for combining outputs from multiple data-driven seasonal forecast models, providing valuable insights into best practices for ensemble forecasting. In addition, ensemble methods, including those based on ML,

can be also more effective in addressing challenges associated with small datasets (Alzubaidi et al., 2023; Safonova et al., 2023; Dietterich, 2000). In this context, it is worth noting that observational data gaps are common in mountainous regions of the Global South (Hock, R. et al., 2019).

90  Water is inextricably intertwined with the development challenges of Central Asia, yet its availability during the growing season remains erratic. The hydrological discharge in Central Asian rivers is subject to large seasonal temperature and precipitation cycles; the latter falls as snow in winter and its melting contributes to spring and summer runoff. The high variability of precipitation during the cold season results in high interannual volatility of river streamflow in the endorheic rivers of Central Asia since most discharge originates from snowmelt in the Pamir and Tian-Shan mountains (Viviroli & 95  Weingartner, 2004). This high hydroclimatic variability underscores the need for improved water availability forecasting during the irrigation season (Xenarios et al., 2019).

Research on seasonal river discharge forecasting in Central Asia can be classified into two mainstream approaches. The first approach explored the predictability of mean discharge from April to September (hereinafter referred to as 'growing season") 100  by using estimates of terrestrial water storage that accumulates in mountain catchments throughout the preceding November to March (hereinafter referred to as "cold season"). Terrestrial water storage in Central Asia is dominated by large annual cycles, with most precipitation during the extended cold season falling from autumn to spring and accumulating as snowpack in the mountain catchments. In the absence of in-situ snow water equivalent (SWE) data, several studies explored the use of proxies such as cumulative precipitation over the cold season (Dixon and Wilby, 2016; Schär et al., 2004), satellite-derived 105  snow cover, antecedent discharge, and other predictors (Apel et al., 2019; Gafurov et al., 2016).

Another approach uses climate indices of global climate oscillations as predictors, some of which are known to have a noticeable impact on hydroclimate variability in Central Asia. It was found that El Niño-Southern Oscillation (ENSO) during its warm phase (aka El-Niño) increases precipitation intensity in Central Asia, most pronounced from autumn to summer 110  (Mariotti, 2007; Chen et al., 2018). In contrast, the cold phase of ENSO (i.e. La-Niña) contributes to below-average precipitation in the region. The Pacific Decadal Oscillation (PDO) can intensify ENSO's effects: during La Niña phase, when the PDO is in its negative phase, Central Asia is more susceptible to severe droughts (Wang et al., 2014). The North Atlantic Oscillation (NAO), Scandinavian pattern (SCAN), and East Atlantic/Western Russia pattern (EAWR), which all are periodic fluctuations in atmospheric pressure between specific regions of the Atlantic Ocean and Eurasia, also affect hydroclimatic 115  variability in Central Asia (Syed et al., 2010). Several studies showed that indices of these climate oscillations can be used to forecast seasonal precipitation (Gerlitz et al., 2019; Umirbekov et al., 2022) and streamflow in the region (Barlow and Tippett, 2008; Dixon and Wilby, 2019).

Another challenge hampering the development of advanced forecasting techniques in the region is a scarcity of in-situ 120  meteorological and hydrological observations, particularly for snow mass measurements. In the past, local hydrometeorological agencies conducted snow depth measurements across the region's main catchments. This practice was discontinued mainly due to the underfinancing of the relevant agencies that persisted for the past three decades (Xenarios et al., 2019). Satellite or reanalysis datasets available in near-real time can be an alternative source for estimating SWE. Still, they might be prone to inherent uncertainties and insufficient spatial resolution to capture variations of accumulated SWE 125  (Mortimer et al., 2020), with larger errors in mountainous regions (Mortimer et al., 2024). Combining multiple satellite-derived or reanalysis estimates may improve snowpack estimation, thereby reducing streamflow prediction uncertainty (Oğulcan Doğan et al., 2023; Mortimer et al., 2020).

This paper tests a new framework for seasonal streamflow forecasting in Central Asia, which combines catchment SWE estimates with climate oscillation indices. In our conceptual framework, basin-averaged SWE represents the initial hydrological conditions, while climate oscillation indices serve as precursors to climate variability during the targeted season. Assuming that precipitation is the dominant driver of streamflow, we incorporate climate oscillations that have a stronger influence on precipitation variability in the targeted basins as additional predictors.Given the region's observational data gaps, this study also evaluates the utility of SWE derived from global reanalysis and satellite products for hydrological forecasting in high-elevated catchments. We used generalised linear regression and machine learning techniques, such as Random Forest, Gaussian Process, and Support Vector Regression, which produce a range of individual forecasts. Finally, we employ an ensemble stacking approach, a type of ensemble learning that uses the predictions from individual models as inputs to a model that produces a more reliable final prediction.

## 2. Study Area

The study area encompasses eight diverse snowmelt-dominated catchments in the Pamir, Hindukush, and Tian-Shan mountains (Figure 1, Table 1). The size of the selected catchments varies from 343 to 296,000 km$^2$, and the mean catchment altitude ranges from 1,700 to 3,500 meters. The catchments include the largest rivers in the region, the Amudarya and Naryn (the main tributary of the Syrdarya), which embed several smaller tributary sub-catchments. For reference, the Figure 1 also depicts the annual precipitation cycles in the study catchments, which in the absence of in-situ measurements were estimated using Climate Terra data. (Abatzoglou et al., 2018). It should be noted that while gridded precipitation products consistently capture annual climatology, they may exhibit high bias in mountainous areas (Hu et al., 2018; Peña-Guerrero et al., 2022).

**Table 1. Major geographical and hydrological characteristics of the study catchments.**

| # | Catchment | Gauging station name | Station location (lat, long) | Catchment area (km$^2$) | Catchment mean altitude (m.a.s.l) | Mean Apr-Sep discharge during 2000-20018 (m$^3$/sec) | Coefficient of variation of discharge during 2000-2018 |
|---|-----------|----------------------|------------------------------|-------------------------|-----------------------------------|-----------------------------------------------------|--------------------------------------------------------|
| 1 | Murghab | Takhta Bazar | 35.96, 62.91 | 35,582 | 1,710 | 41 | 0.49 |
| 2 | Amudarya | Kerki | 37.84, 65.23 | 296,300 | 2,550 | 1,876 | 0.29 |
| 3 | Varzob | Dagana | 38.70, 68.79 | 1,279 | 2,700 | 79 | 0.23 |
| 4 | Vaksh | Komsomolabad | 38.86, 69.94 | 28,908 | 3,530 | 996 | 0.1 |
| 5 | Kashkadarya | Varganza | 40.81, 73.26 | 343 | 2,663 | 18 | 0.35 |
| 6 | Zarafshan | Dupuli | 39.49, 67.80 | 10,310 | 3,125 | 243 | 0.17 |
| 7 | Naryn | Toktogul | 41.77, 73.29 | 46,667 | 2,940 | 561 | 0.22 |
| 8 | Chu | Kochkor | 42.25, 75.83 | 5,305 | 2,934 | 35 | 0.31 |

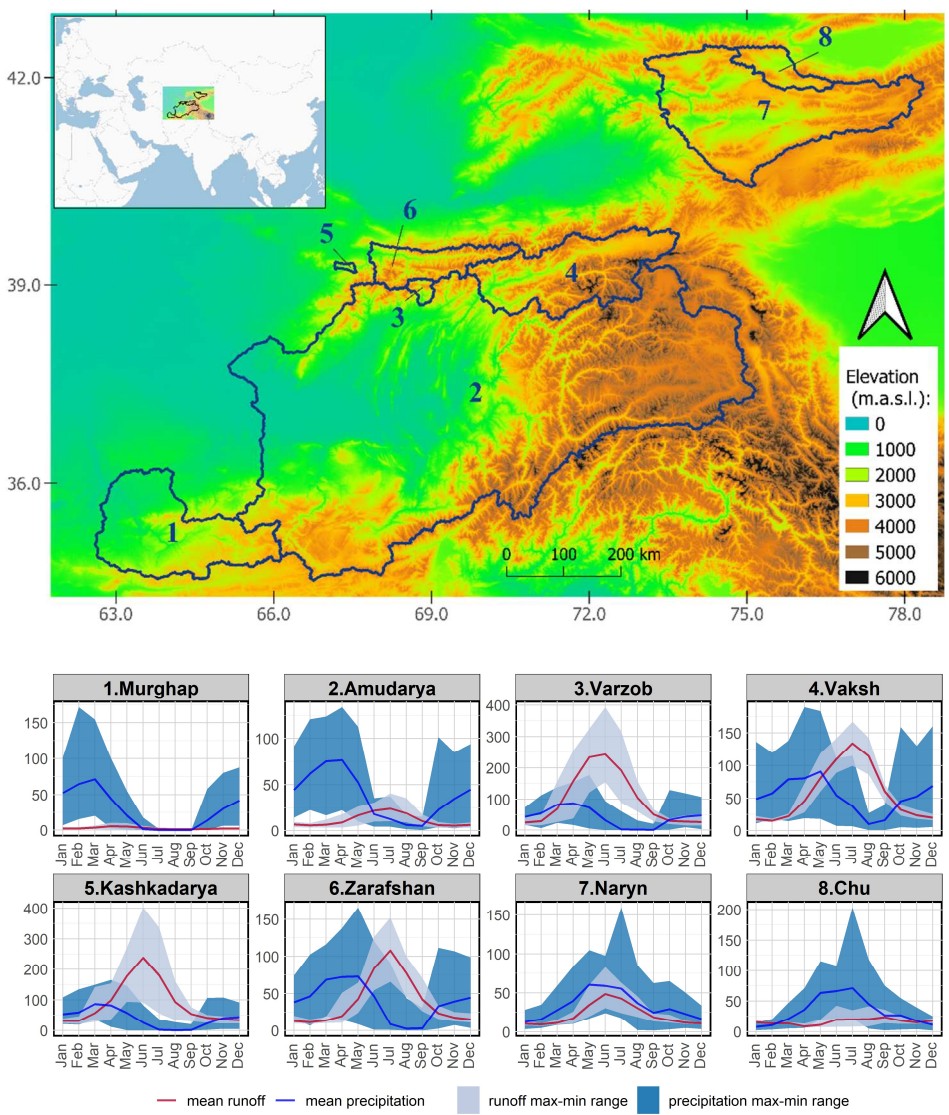

**Figure 1. Location of the study catchments (upper map), monthly means and ranges for precipitation and runoff for 2000-2018 in mm (bottom graphs).**

### 3. Data

155 The predictand variable represents seasonal discharge, calculated as the mean discharge from April to September. We obtained monthly discharge data for the study catchments from 2000 to 2018 from hydrometeorological agencies in Central Asian countries. We aggregated these into mean discharge from April to September, resulting in approximately eighteen observations of seasonal discharge for each catchment.

160 Operational seasonal streamflow forecasting in snowmelt-dominated catchments primarily relies on snow measurements but usually also incorporates additional variables that reflect initial hydrological conditions, such as accumulated precipitation, antecedent flow, and basin soil moisture. Due to the limited availability of streamflow observations and the overall objective of assessing the added value of teleconnections compared to snowpack-only predictions, we restricted the predictors to two types: SWE and climate oscillation indices.

165

As the primary predictor variable, we use four basin-averaged SWE estimates that can be derived from global climate datasets available in near real-time (Table 2). These include two SWE estimates from global and regional reanalysis datasets, i.e. ERA5-Land (Muñoz-Sabater et al., 2021) and Land Data Assimilation System Central Asia (McNally et al., 2022). In addition, we

obtained two SWE estimates using the GEMS snow model (Umirbekov et al., 2023) forced by global precipitation and temperature data available in near-real time. One simulated SWE time series is obtained by forcing the snow model with the Multi-Source Weather dataset, generated by bias-correcting and downscaling ERA5 (Beck et al., 2021). The fourth SWE estimate is simulated using precipitation estimates from the Integrated Multi-satellite Retrievals GPM IMERG v6 (Huffman et al., 2019) and temperature estimates from MSWX. We used a 'Late Run' version of GPM IMERG precipitation estimates, accessible in near-real time albeit lacking adjustments using ground precipitation data as in the 'Final' product, which becomes available two months later.

Candidates for additional predictors include the monthly indices of the El Niño–Southern Oscillation (ENSO), the Pacific Decadal Oscillation (PDO), the North Atlantic Oscillation (NAO) and the Scandinavian Pattern (SCAN).

**Table 2. Snow water equivalent estimates and climate oscillation indices that were used as predictors in this study**

| Type | Predictor (abbreviation) | Description | Source |
|---|---|---|---|
| Snow Water Equivalent estimates | ERA5-L | Retrieved from the ERA5-Land reanalysis dataset | Muñoz-Sabater et al., 2021 |
| | FLDAS | Retrieved from the Land Data Assimilation System Central Asia | McNally et al., 2022 |
| | MSWX | Simulated using GEMS model forced by precipitation and temperature estimates from MSWX dataset | Beck et al., 2021 |
| | GPM | Simulated using GEMS model forced by GPM IMERG precipitation and MSWX temperature | Huffman et al., 2019 |
| Climate Oscillation Indices | SOI | Southern Oscillation Index | Ropelewski and Jones 1987 |
| | PDO | Pacific Decadal Oscillation | Mantua et al 1997 |
| | EAWR | East Atlantic/West Russia pattern (EAWR) | Barnston and Livezey 1987 |
| | NAO | North Atlantic Oscillation (NAO) | Barnston and Livezey 1987 |
| | SCAN | Scandinavian pattern | Barnston and Livezey 1987 |

The hydrological dataset used in this study comprises 18 seasonal discharge observations per study catchment, spanning 2000 to 2018. This highlights the data availability limitations in the region. To address the challenges posed by this small dataset, we employed the aforementioned approach, which integrates multiple diverse estimates of SWE and utilises an ensemble methodology described below in section 4. "Methods". It is worth noting that historical data for most catchments also spans from the 1970s to the 1990s, with relatively more complete records available for the largest rivers extending up to 2000. However, incorporating these earlier records is challenging, as the datasets used to derive some SWE estimates (FLDAS and GPM) are only available from 2000 onward. Extending the observations back in time would restrict the ensemble to only ERA5-L and MSWX datasets, thereby reducing its diversity and potentially compromising its robustness. In addition, using older discharge records in our framework may be problematic due to the non-stationarity of climate and hydrological systems (Pagano and Garen, 2005; Livneh and Badger, 2020), as well runoff alterations in some large basins induced by land-use changes over the past century (Hou et al., 2023).

## 4. Methods

### 4.1 Determining associations between climate oscillations and hydroclimatic variability across study catchments

To determine linkages between the selected climate oscillations and hydroclimatic variability across the catchments, we calculated Spearman's rank correlations with precipitation during months with higher magnitude and interannual variability. We used the global TerraClimate precipitation dataset (Abatzoglou et al., 2018) to construct catchment-averaged precipitation time series from 1979 to 2020. The annual precipitation cycle in the studied catchments exhibits two distinct sub-regional patterns (see Figure 1). Catchments in the Pamir and western Tian-Shan experience increasing precipitation during winter, peaking in the spring, and decreasing during summer. In contrast, the Naryn and Chu catchments, located in the interior and northern Tian-Shan, receive most precipitation from late spring to early summer and less precipitation in winter. Across all catchments, the interannual variability is greatest during the months with the highest precipitation totals. We have defined the standard peak precipitation season for the region as February to June since this period covers the months with the highest precipitation amounts and the greatest interannual variability across all the studied catchments. We then calculated Spearman's rank correlation coefficient between the catchment averaged precipitation for February-July (referred to here as 'peak precipitation season') and each climate oscillation index at varied lead-lag times. To identify when oscillations show the strongest association with the precipitation season, we calculated the correlation for each oscillation index from August of the preceding year to July, the final month of the peak precipitation season. In addition, we computed correlations between the climate indices and mean discharge during the growing season using the same procedure.

### 4.2 Stacked ensemble-based prediction of seasonal discharge

Our modelling framework employs an ensemble stacking approach. This machine learning technique combines predictions from multiple base models and uses them as inputs to a higher-level meta-learner, also known as stacking model. This approach has seen increasing application in the hydrological field in recent years (Zounemat-Kermani et al., 2021), including long-term streamflow forecasting, drought monitoring, and real-time flood forecasting (Mallick et al., 2022; Granata and Di Nunno, 2024; Li et al., 2019; Xu et al., 2024).

The ensemble stacking workflow consists of four main steps depicted in Figure 2. In the first stage, we combine each basin-averaged SWE estimate with climate oscillation indices at months when they exhibit higher association with in-season precipitation peaks. Four different SWE products result in four datasets with varying SWE estimates but the same set of selected climate oscillation indices for each catchment. Any set of predictors for each basin includes a maximum of three variables: one SWE estimate and up to two climate oscillation indices.

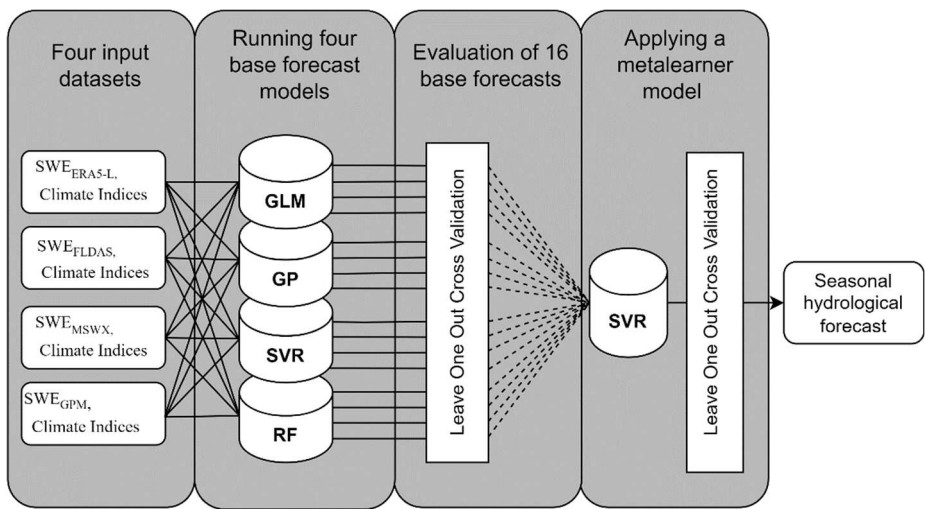

**Figure 2. Workflow of the ensemble-based forecast approach**

We then use four different forecast models (from now on referred to as "*base models*"), each forced with the four input datasets to produce a range of 16 seasonal predictions. The four base models are comprised of the generalised linear model (GLM)
with Gaussian identity link, gaussian process regression with the linear kernel (GP), support vector regression (SVR) with the linear kernel (SVR) and random forest (RF). The latter two model algorithms have parameters that control internal model complexity. For example, the "*cost*" parameter in SVR limits training errors against maximising the margin of the decision function, and "*mtry*" in RF determines the number of predictors that can be taken into account at each split point of a single tree.  We confine these parameters to relatively lower levels (cost=0.3 in the case of SVR and mtry=1 in the case of RF), which
helps to avoid overfitting and facilitates a higher degree of generalizability (Najafi and Moradkhani, 2016; Safonova et al., 2023).

In the next step, we evaluate each of the 16 base model predictions using leave-one-out cross-validation (LOOCV), which is well-suited for the small dataset context. It is worth noting that LOOCV is a standard practice in developing and evaluating
water supply forecasting models in the western U.S. (Fleming and Garen, 2022). For each base model, we compute a LOO Cross-Validated $R^2$ value based on its deterministic hindcasts. Rather than using all 16 base model hindcasts in subsequent steps, we apply a selection threshold: only base models with an LOOCV $R^2$ greater than 0.2 are retained for further analysis. This threshold requires a leave-one-out cross-validated R-squared coefficient of base model performance to be greater than 0.2 to be considered for further analysis. This threshold was optimal during LOOCV in terms of predictive performance for
the stacking ensemble.

In the final step, those base model predictions that pass the LOOCV test become inputs for a final forecast model (from now on called "*meta-learner model*"). Since all selected base model predictions would exhibit some degree of correlation among themselves, we employ the SVR algorithm as a meta-learner model, which is known to be less sensitive to multicollinearity
(Farrell et al., 2019). The final prediction of the meta-learner model is again assessed using LOOCV.

We apply the procedures described above for each standard forecast issue time adopted by hydrological agencies in Central Asia, starting from January 1st, that is a three-month lead time concerning the April-September season, and ending with the final forecast issued just before the start of the season, i.e., on April 1st.  Each forecast uses inputs that are accessible by its
issue date. For example, three-month lead forecasts can only use estimates of catchment SWE by January 1st and state of climate oscillations in previous months. To attain parsimonious forecast models, rather than incorporating all studied climate oscillation indices into a set of predictors, we followed a stepwise approach: each of the climate indices was added one at a

time to the predictors set, which was then evaluated. This approach led to a final predictor set with the minimal combination necessary to produce plausible predictions for each catchment and each forecast issue date.


The overall modelling framework employed in this study prioritizes parsimony by relying on relatively simple type of models with few or no internal parameters and a maximum of three input variables. This design minimizes the number of parameters to estimate from the limited sample size, making the approach particularly well-suited for short datasets.

### 4.3 Uncertainty estimation


We applied bootstrapping (using 500 bootstrap samples) to assess predictive uncertainty by resampling the input data and retraining both the base models and the meta-learner on each bootstrapped sample. From the ensemble of bootstrapped predictions, we derived 80% prediction intervals, defined by the 10th and 90th percentiles.


### 4.4 Determination of supplementary importance of incorporating climate oscillations as additional predictors

We implement two track evaluation analyses to determine the value of adding COs as additional predictors into snow-based forecasts. First, we elaborate forecast models that use only SWE estimates as predictors, using the same approach described in the previous section, and compare their performance with those that use both SWE and COs. Second, we determine the relative

importance of COs using the *feature importance ranking measure* method (Greenwell et al., 2018), which quantifies how much each input variable influences the predictions made by the model. The method assesses the impact of each input variable by estimating partial dependence plots (Friedman, 2001) and assigning higher (lower) importance rank to features that exhibit a steeper (flatter) partial dependence effect.

## 5. Results


### 5.1 Evaluation of SWE estimates

Figure 3 summarises the correlation coefficients between catchment-averaged SWE at different forecast issue dates and mean discharge during the growing season. The SWE estimates obtained from global reanalysis and satellite data exhibit varied degrees of connection with the seasonal discharge. For all catchments, the correlation in general tends to increase with shorter lead times, i.e., with SWE estimates for January 1st having the lowest correlation and those for April 1st having the highest.

While SWE estimates based on ERA5-L and MSWX generally show a higher correlation with seasonal discharge across most catchments, in the absence of in-situ snow measurements, it is impossible to assert which of the four SWE estimates is relatively more consistent.

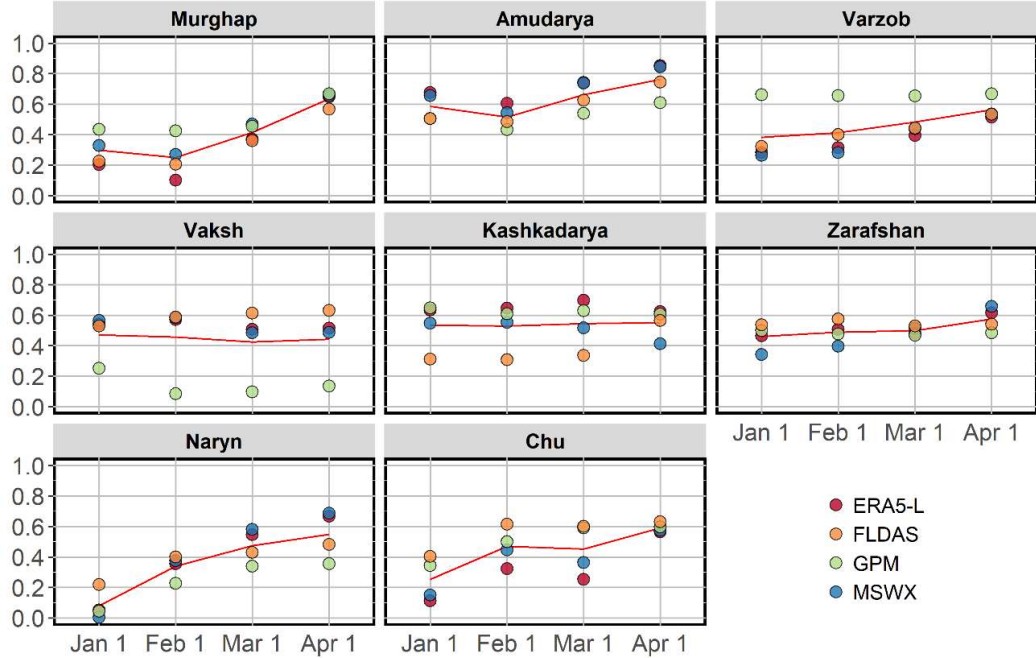

**Figure 3. Pearson`s correlation coefficients between the SWE estimates and mean seasonal discharge between April and September at different forecast lead months.** The red line is the median across all snow products.

### 5.2 Association between climate oscillations and hydroclimatic variability across the study catchments

Evaluation of the climate oscillation indices revealed diverse associations with peak season precipitation and mean river discharge during the growing season across the catchments (Figure 4, upper graph). In all catchments, the February-July precipitation exhibits a robust and persistent association with ENSO, represented by the Southern Oscillation Index (SOI) and the Pacific Decadal Oscillation (PDO), over an extended timeframe compared to other oscillations. A significant negative correlation exists between peak precipitation season and SOI in all catchments, evident three months before the season's commencement. This relationship persists for a longer duration compared to any other climate oscillation. On the other hand, PDO exhibits a positive link with seasonal precipitation, becoming noticeable as early as four months before the season's onset and reaching its most substantial level in November. The selected lead months of SOI and PDO exhibit a higher correlation, possibly because the latter also mirrors the ENSO phenomenon.

Like ENSO and PDO, the East Atlantic/West Russia pattern (EAWR) consistently demonstrates a stronger correlation across most catchments before the peak precipitation season. Notably, the October state of EAWR shows a substantial positive correlation with peak precipitation across all catchments; however, it becomes more variable as the season progresses.

On the other hand, the North Atlantic Oscillation (NAO) and the Scandinavian Pattern (SCAN) show a relatively less pronounced association with the peak precipitation season, with correlations that vary depending on the lead time. From December to March, the NAO shows a weak but persistent negative relationship with the peak precipitation season in most catchments. In January, at the beginning of the peak precipitation season, a considerable portion of the catchments demonstrates a negative correlation with the state of SCAN. However, as the season progresses and reaches March, there is a noticeable shift, with all catchments showing a stronger and more positive correlation with the state of SCAN.

315 The correlation between the climate indices and mean river discharge during April-September exhibits almost the same pattern (Figure S1 in the Supplement). This implies that interannual discharge variability is predominantly driven by precipitation between February and July.

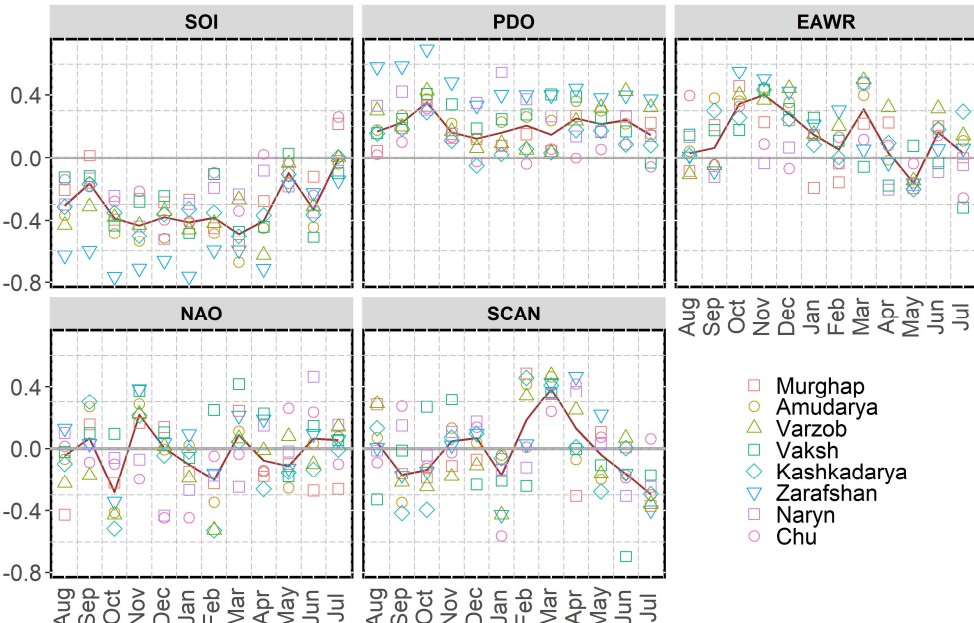

320 **Figure 4. Spearman's rank correlation coefficients between the climate oscillation indices and precipitation from February to July precipitation.** The X-axis denotes months of a climate index. The red line represents a median for correlation coefficients across all catchments in each month.

### 5.3 Performance of seasonal discharge forecasts

325 Figure 5 below summarises a set of final predictors per studied catchment, obtained after screening COs associations with peak precipitation and mean discharge during the growing season and following a stepwise selection procedure using the ensemble-stacking forecast approach described in section 4.2.

While the input dataset for the base models included SWE estimates, the combination of climate oscillations they rely on varies

330 depending on a catchment location and elevation. In most catchments, there is a higher correlation between the late autumn state of PDO and the winter state of SOI. To avoid redundancy and potential issues with multicollinearity, we did not include both indices as predictors in the models for the same basin. Instead, PDO or SOI was selected for each basin's model based on which index exhibited a stronger predictive relationship. As a result, SOI mostly appears as a predictor in Tian-Shan catchments, and PDO generally persists as a predictor in catchments located in Pamir. The winter state of NAO and SCAN are

335 another source of predictability in many of the catchments but have variable temporal signatures. In the case of the Murghap, where workable base models were obtained only for the April 1st forecast, they rely solely on SWE estimates and SCAN as predictors.

Selected climate indices tend to have the same temporal lags for neighbouring catchments. For instance, the Naryn and Chu

340 catchments in the Tian-Shan, which have similar seasonal precipitation patterns, use the NAO condition in January as one of their predictors. The Varzob and Zarafshan rivers, both high-elevation tributaries of the Amudarya, use the January state of SCAN.

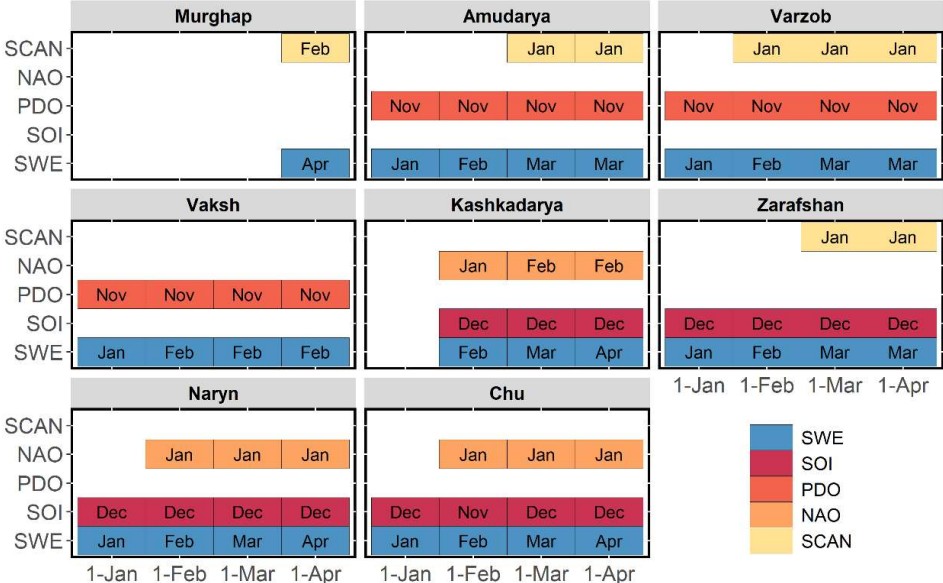

**Figure 5. Predictors at different forecast issue times.** Abbreviations within boxes indicate the month of the respective climate oscillation index or catchment-averaged SWE used as predictors. For example, the April $1^{st}$ forecast models for the Amudarya use as predictors the SWE estimate as of the beginning of March, the state of the PDO index in November, and the SCAN index in January.

The ensemble-based forecasting framework plausibly simulated seasonal discharge across all catchments, albeit with varying temporal performance based on lead time (Figure 6). The meta-learner model's LOOCV $R^2$ coefficient varies between 0.2 and 0.5 for the extended lead time forecast (1 January). It gradually increases with decreasing lead time, surpassing 0.9 for the April 1 forecast for most catchments except Murghap and Varzob. The accuracy of the meta-learner model forecasts depends on the number and diversity of the resultant individual base models, which is superior to those of the latter. This underscores the strength of ensemble approaches, which outperforms single-model approaches, as demonstrated in similar studies (Hagedorn et al., 2005; Najafi and Moradkhani, 2016; Fleming et al., 2021).

Due to the threshold criterion ($R^2>0.2$) for a base model to be included in the final meta-learner prediction, the resulting stacked ensembles typically consist of fewer than 16 base models. We observe two trends in this regard: (1) the later the issue date, the greater the number of base models included in the ensemble, and (2) larger catchments tend to incorporate more base models for certain rivers, such as the Varzob, Kashkadarya and Chu, this results in fewer base models being used for ensemble stacking. For the Murghap and Kashkadarya catchments, no feasible base models were obtained for the January $1^{st}$ forecast. Furthermore, workable base models and the derived meta-learner model for Murghap are only obtainable for the April $1^{st}$ forecast. Table S3 in the Supplementary Material provides information on number of base models used for stacking the meta-learner model per each basin and forecast issue date.

No model types are consistently superior in accuracy across all lead times, especially for the final (April $1^{st}$) forecast. However, the base models' performance has some distinct spatial heterogeneity, depending on which SWE product they use. For example, all base models for the Vaksh retained after cross-validation rely mostly on $SWE_{ERA5-L}$ or $SWE_{MSWX}$ as inputs. In contrast, forecasts for the Varzob catchment have more base models using $SWE_{FLDAS}$ and $SWE_{GPM}$. The seasonal discharge in the largest catchments, such as Amudarya and Naryn, is also better explained by base models that use $SWE_{ERA5-L}$ or $SWE_{MSWX}$.

The results suggest that models incorporating GPM IMERG have lower predictive accuracy, reflected in overall lower cross-validation performance, except in the highly elevated Varzob and Zarafshan catchments. This is likely due to the lower

accuracy of the GPM IMERG`s Late Run product, which includes only climatological adjustment. In contrast, its final product

("Final Run") comes with adjustments using gauge data. However, the latter is only available at a three-month latency, precluding its operational forecasting use.

We tested several other ML techniques as base models, including using the same models with non-linear kernels. In most cases, the presented combination of models yielded a better accuracy regarding MAE and R-squared coefficients during

LOOCV. Sometimes, certain non-linear models produced slightly better predictions depending on the basin or issue date. However, the existing structure still showed superior accuracy when generalising across all basins and issue dates. We assume this may be due to two major and non-exclusive factors: (1) fewer observations and predictors, which makes non-linear machine learning models less efficient and prone to overfitting, and (2) the selection of predictors based on a linear metric (Pearson's correlation) may have inherently favoured linear models.

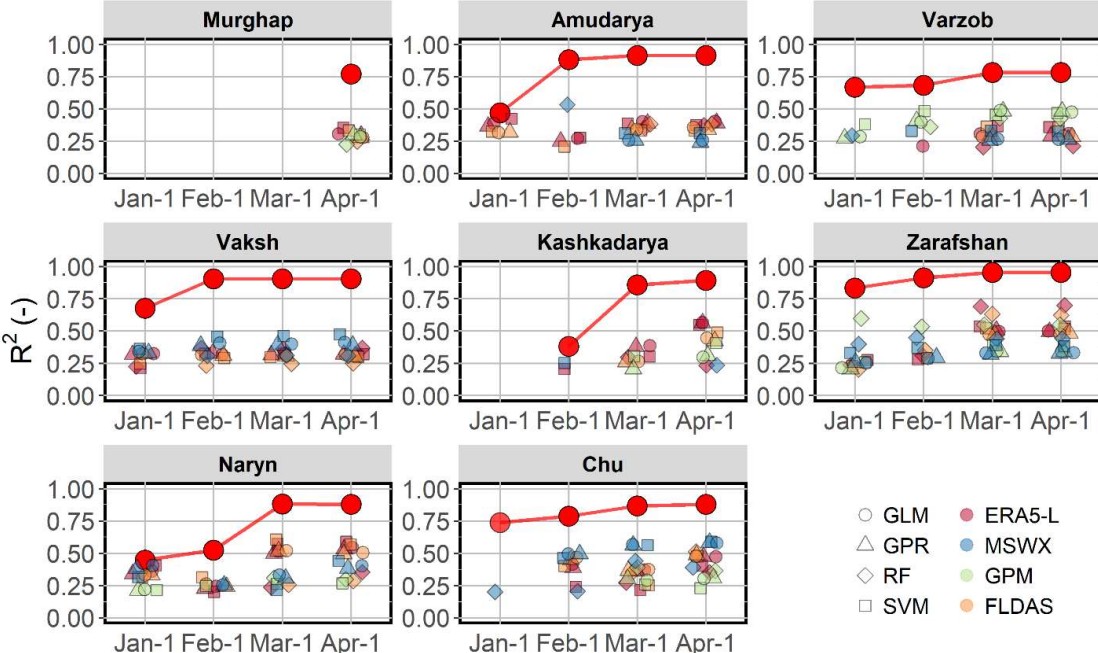


**Figure 6. LOOCV R-squared coefficients of individual base models at different lead months and the LOOCV R-squared of the meta-learner model (red line).**

The inclusion of climate indices generally enhances forecast accuracy across most catchments, as reflected in the generally lower normalized MAEs for models that combine SWE and climate indices compared to those that use only SWE (Figure 7).

However, there are exceptions, such as in the February forecast for Kashkadarya and the January forecast for Vaksh, where SWE-only models exhibit lower errors. The improvement from including climate indices is particularly evident in catchments situated in the Tian-Shan mountains, such as Naryn and Chu, where SWE-only forecasts result in substantially higher errors. Moreover, the difference in MAEs between the two model types becomes more pronounced with reduced lead times in these catchments. A similar pattern is observed in the high-elevation Zarafshan catchment. In contrast, in Amudarya and Murghap,

large and relatively low-elevation catchments located in the Pamir region, the incremental differences between the two model types for the April 1 forecast are minor or absent, suggesting that the inclusion of climate indices provides limited added value in these cases.

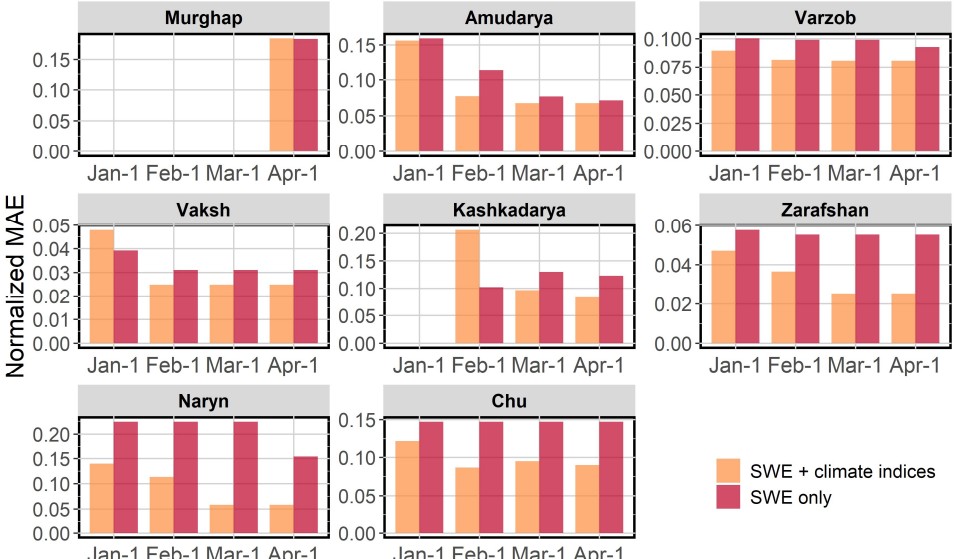

**Figure 7. Normalized MAE of the meta-learner hindcasts at different forecast lead months**


Comparison of observed and predicted seasonal discharge across basins and forecast issue dates (**Figure 8**) showcases the performance and limitations of the modelling framework. The alignment of hindcasts with observed discharge in most basins indicates reasonable predictive skill, though deviations are sometimes evident. Hindcasts initialised earlier, such as on January 1st, tend to show a larger scatter, highlighting higher uncertainty than forecasts initialised closer to the target season (e.g.,

March 1st or April 1st), which display tighter clustering around the 1:1 line. Basins such as Murghap and Kashkadarya exhibit greater scatter in predictions across all issue dates, particularly at the extremes, reflecting challenges in low-elevated basins. For the largest basins like Naryn and Amudarya, systematic biases are also evident in early issue date hindcasts, with overestimation at lower and underestimation at higher quantiles. These biases likely stem from the uncertainties of snowmelt and hydrological dynamics in these large basins, where higher spatial variability in snow accumulation complicates

predictions. In contrast, the Zarafshan, Varzob, and Chu basins align better between hindcasts and observations across all initialisation dates, suggesting more predictable hydrological responses.

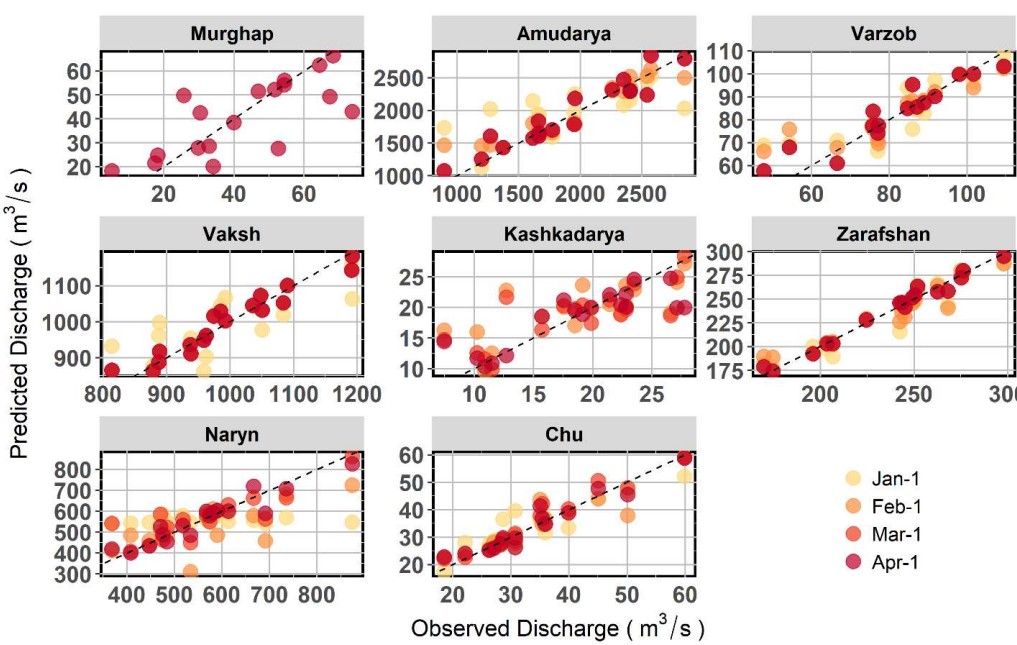

**Figure 8. Observed seasonal discharge and hindcasts produced by meta-learner models at different lead months**

## 5.4 Predictive uncertainty

Interannual streamflow variability was generally well captured by the meta-learner models, with bootstrap-based 80% predictive uncertainty closely aligning with observations (Figure 9). However, the predictive uncertainty varies across basins, reflecting basin-specific characteristics that influence predictive reliability. Hindcasts initialised earlier, such as January 1st, tend to have broader uncertainty bounds and greater variability. In contrast, as forecast issue dates approach the target season, they become more consistent, with narrower uncertainty bounds and better alignment with observations. Hindcasts for basins with smaller catchment areas or/and located at lower elevations, such as Kashkadarya, tend to have wider uncertainty bounds. Similarly, Murghap, a basin with lower seasonal discharge, shows higher deviations in predictions. For these two basins, higher predictive uncertainties may be also be attributed to the comparably higher interannual variability of the seasonal discharge (Table 1). In larger basins like Naryn and Amudarya, variability in uncertainty bounds is more pronounced for early issue dates. However, hindcasts for these basins become more relatively less uncertain with later initialisation dates as the models incorporate updated snowpack data, narrowing the uncertainty bounds. In contrast, Zarafshan and Chu, high-altitude basins, demonstrate stable predictions and tight uncertainty bounds across all initialisation dates. The hindcasts exhibit higher predictive uncertainty for some catchments in certain years, potentially due to extreme climate conditions or unusual snowpack accumulation patterns. For instance, the hindcast tend to have relatively higher uncertainty in the Amudarya and Vaksh basins in 2008, a year that recorded one of the lowest terrestrial water storage levels in those catchments (Gafurov et al., 2024). Furthermore, although they correctly guessed the trend during those years, the 80% predictive uncertainty intervals for the small, high-elevated Varzob catchment fell short of capturing the two lowest observed streamflow values.

It should be noted that the uncertainty intervals are estimated by bootstrapping a relatively short streamflow time series and do not account for uncertainty caused by the potential limited representativeness of the actual natural variability by the observations.

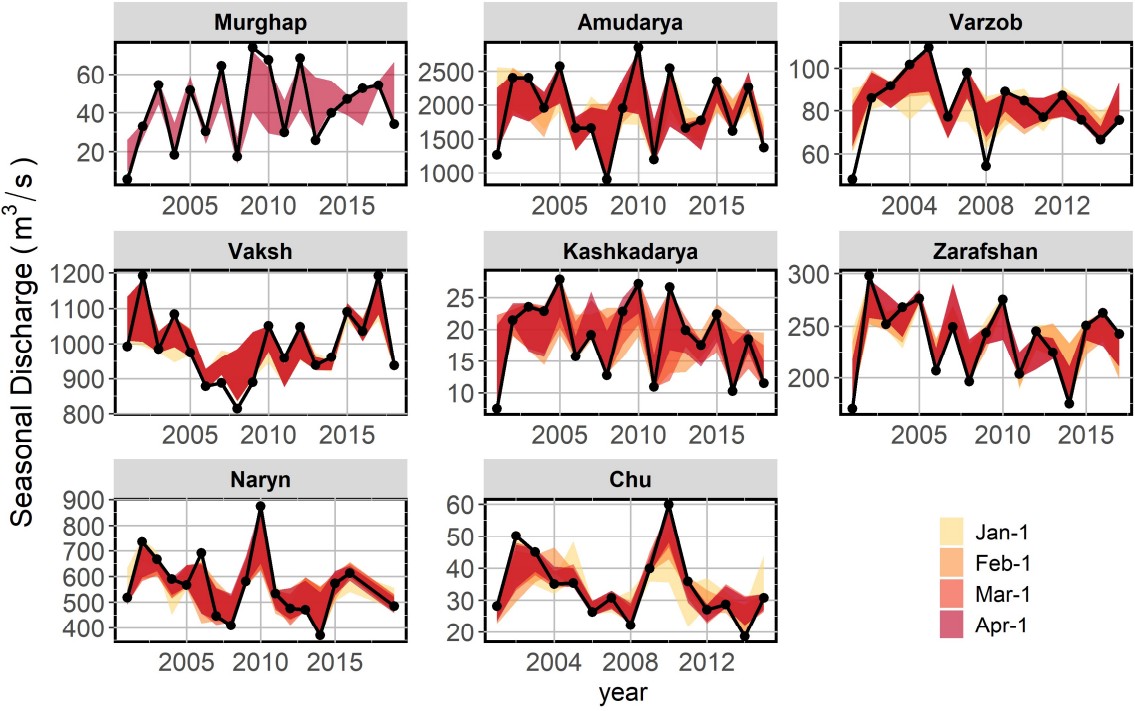

**Figure 9. The 80% predictive uncertainty intervals of the meta-learner models at different lead months**

### 5.5 Importance of climate oscillation indices as predictors

The importance of predictors varies depending on the catchment location and the forecast issue date (Figure 10). Regardless of the forecast issue date, SWE is a major predictor in most catchments located in Pamir, and its significance generally arises with decreasing lead times. Its incremental value is evident in the basins in the western part of the study area, the Pamir Mountains. Nevertheless, the supplementary predictive value of COs is visible in all basins regardless of their location except for Murghap, where integration of COs does not noticeably improve prediction accuracy compared to models relying only on SWE. The predictive power of COs is highest for the two catchments located in the inner and northern Tian-Shan, Naryn and Chu. Especially in Chu, the COs contribute to more than half of the predictive power of the forecast models across all forecast issue dates. In addition, a higher reliance on COs is also evident in the high-elevation catchments of Zarafshan and Varzob, with their importance surprisingly increasing at later forecast issue dates.

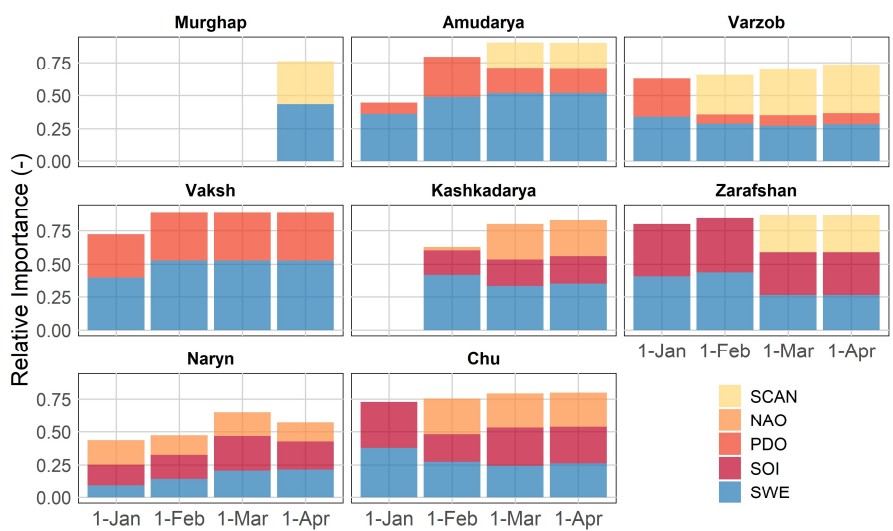

**Figure 10. The relative importance of predictors at different forecast issue dates**

### 6. Summary and Discussion

Our findings suggest that valid SWE estimates, suitable for operational seasonal river discharge forecasting, can be effectively derived from global reanalysis or satellite data. Still, they are subject to spatial bias and uncertainty, which may be due to uncertainties in underlying precipitation and temperature inputs. The uncertainties in the SWE estimates may propagate across time and become more pronounced during the snow ablation phase by the end of the cold season. Combining SWE data from multiple global sources helps mitigate these biases, and predictions that pass cross-validation filters reflect the accuracy of SWE products specific to catchment locations. Nevertheless, although catchment-averaged SWE estimates improve with the assimilation of multiple snow products, they may still tend to contain spatial uncertainties that increase during the ablation phase.

Multiple global ocean-atmospheric oscillations modulate the seasonal hydroclimatic patterns in the Pamir and Tian-Shan mountains, each with different temporal effects. The findings suggest that the magnitude of both seasonal precipitation and discharge is associated with the late autumn to winter state of ENSO (approximated in our study with SOI). PDO is known to mimic ENSO-like variability on monthly to annual scales and has a pronounced impact on the interdecadal scale (Zhang et al., 1997). This could explain the similarity in dominant lead times observed in our analysis with SOI. Late winter to spring states of NAO and SCAN contribute to hydroclimatic predictability in many studied catchments, mainly showing higher significance in the Tian-Shan domain. All these spatial and temporal patterns are broadly consistent with several earlier findings (e.g., Mariotti 2007, Wang et al. 2014, Dixon and Wilby 2019, Gerlitz et al. 2019).

The associations between the climate indices with precipitation and discharge exhibit an almost identical pattern, implying that interannual volatility of streamflow during the growing season is substantially driven by the peak precipitation period, which we determine as February to July. This implies that SWE accumulated by the middle of winter is a weak precursor of hydrologic variability in the upcoming season, which our findings assert. On the other hand, this serves as an argument for using climate oscillation indices beside the catchment snowpack in discharge forecasts at extended lead times. Following the

traditional approach towards seasonal hydrological predictions, SWE estimates initial hydrological conditions and climate oscillation indices as a proxy of climate variability during the target season.

Our experiment confirms this by demonstrating the complementary roles of SWE and climate oscillation indices in improving discharge hindcasts at extended lead times. The resulting forecast models generate credible hindcasts of seasonal discharge across all studied catchments, albeit with performance variations depending on lead time. The forecast models incorporating both SWE and COs perform better than the SWE-only models, evidenced by lower forecast errors. The resulting forecast models underscore the significance of SWE as one of the primary predictors in most catchments in the Pamir region, with its importance becoming more pronounced during the peak SWE period, typically occurring in mid-spring. Nevertheless, the forecast models gain valuable predictive power from climate oscillation indices during extended and shorter lead times, but the importance of specific climate oscillations as predictors varied across catchments. In most catchments, the SOI, PDO, or both were utilised, indicating the dominant influence of ENSO and other climate variability patterns in the Pacific Ocean. Moreover, the results suggest that the NAO and SCAN exhibit a relatively higher predictive power for catchments in the Tian-Shan region.

The predictive importance of climate oscillations equalled or exceeded that of SWE in the Naryn and Chu catchments located in Tian-Shan, as well as in high-elevated catchments in Pamir, such as Zarafshan, Varzob, and Vaksh. The former might be explained by a distinctive precipitation cycle across Tian-Shan, which peaks during summer, i.e., considerably later than the final forecast issue date (April 1$^{st}$). Consequently, SWE estimates have comparably smaller power to capture upcoming hydroclimatic variability than other catchments where precipitation peaking occurs during spring months and thus is embedded in SWE estimates by April 1$^{st}$. This contrasts with the forecast model for the Murghap catchment, which doesn't integrate any climate oscillations, as precipitation in this catchment peaks before spring. The higher predictive power of the oscillations for high-elevated catchments may be attributed to the poorer performance of the satellites and reanalyses of precipitation estimates over high elevations in the region (Peña-Guerrero et al., 2022), which subsequently propagate as uncertainties in the SWE estimates. In this regard, the higher predictive performance of climate oscillation indices across those catchments is likely because they compensate for errors in SWE estimates.

Based on the abovementioned, we identify three specific cases when the incorporation of COs as additional predictors helps to improve seasonal discharge forecasts in snow-dominated catchments:

1) *Extended lead time forecasts with early seasonal SWE*: When seasonal discharge forecasts are made well in advance, but SWE is not a reliable representation of seasonal terrestrial water storage, climate oscillations may provide additional insights into anticipated hydroclimatic conditions.

2) *Dominant climate variability regime during the target season:* When the seasonal discharge is more influenced by in-season climate variability than by accumulated SWE before the season, climate oscillations can serve as adequate proxies for this variability.

3) *Uncertainties in catchment SWE estimates:* High uncertainties in SWE estimates for a particular catchment result in higher errors in discharge predictions. These uncertainties can be partially compensated by leveraging the forecasts with climate oscillations, leading to more accurate seasonal discharge predictions.

In-situ observations of essential climate variables, such as snowpack properties, are especially scarce in mountainous regions of the Global South, impeding hydrological forecasting. Previous research has demonstrated how, in the absence of in-situ snow observations, satellite-derived snow cover, precipitation and temperature can serve as proxies of terrestrial water storage and improve seasonal discharge forecasting in Central Asia (Apel et al., 2018; Gafurov et al., 2016). Additionally, other studies

have investigated how climate indices characterize hydroclimatic variability in the region over longer lead times (Dixon and Wilby, 2019). By combining the strengths of these two approaches, our modelling framework offers a new way to make hydrological predictions in the region. It leverages an ensemble technique that uses multiple estimates from global data, a diverse set of more straightforward types of machine learning methods with loose tuning parameters. These elements allow us to achieve plausible forecast models even when in-situ discharge observations are short.

## Code and Data Availability

R script to reproduce results in this paper is available at Zenodo (Umirbekov et al., 2025), under the Creative Commons Attribution CC BY 4.0 International license. The same record contains related data files, including mean seasonal discharge of the studied catchments, gridded daily SWE data for the study area, and monthly indices of the climate oscillations.

## Author contributions

AU and DM designed the study. All authors evaluated the findings and contributed to the interpretation of results. All authors contributed to writing and reviewing the manuscript. DM supervised the project.

## Competing interests

The contact author has declared that none of the authors has any competing interests.

## Acknowledgements

We would like to thank the two anonymous reviewers for their valuable comments and suggestions, which helped improve the manuscript. We extend our sincere appreciation to the open-source developer community and individuals behind numerous R packages (R Core Team, 2025), including but not limited to caret (Kuhn, 2008), terra (Hijmans, 2023), vip (Greenwell and Boehmke, 2020), and dplyr (Wickham et al., 2023).

## Financial Support

This research has been supported by the Volkswagen Foundation (grant no. 96 264).

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
