# Peer review of "The Value of Hydroclimatic Teleconnections for Snow-based Seasonal Streamflow Forecasting in Central Asia"

_Hydrology and Earth System Sciences, 2024_

## Author Comment (AC1)

Dear Reviewer,

Thank you for your valuable and constructive comments on our manuscript. We have carefully considered all the suggestions. We will expand the literature review to provide a broader context by referencing relevant work from North America and other regions, especially focusing on similar hydroclimatic and water supply forecasting studies. We will also refine the terminology to clarify distinctions between process-based and data-driven approaches and ensure the manuscript uses consistent, field-appropriate language. Additionally, we will revise the methodological descriptions to better explain the rationale behind the chosen models and improve the clarity and precision of our wording throughout the paper. We believe that the revised version will be more comprehensive and better aligned with current research in the field. Please find below your referee comments (in black) and our responses (in blue).

With regards,

The authors.

Review of 'The value of hydroclimatic teleconnections for snow-based seasonal streamflow forecasting,' Umirbekov et al., HESS Discussions

This submission summarizes the motivation for, and development, implementation, and performance of, a data-driven model for seasonal river runoff volume forecasting in Central Asia. The method uses as predictors a combination of SWE data from existing large-scale operational remote sensing and land surface modeling products, and indices of various atmosphere-ocean circulation patterns, as predictors; it employs a multi-model ensemble model structure, which has some machine learning elements; and is intended to serve as an actual operational forecasting tool for directly supporting water management decision-making.

My overall recommendation is for publication pending minor revisions.

The article will be an excellent contribution to HESS. The paper is succinct and well-organized. The study location considered has been historically understudied, though admittedly not so severely as some other regions of the developing world. The candidate predictor data combinations are geophysically sensible, but also to some degree original in the particular way they are used here. Though it is possible to quibble with certain aspects of the predictor selection process used, it is a reasonable and defensible approach for the task at hand. Its multi-model ensemble philosophy is fully consistent with a large body of evidence demonstrating its value, yet it is only one of a tiny handful of hydrologic modeling examples where several separate data-driven/statistical/machine learning modeling systems are used and their results

pooled to form a best estimate, and it is also novel as implemented here. Unlike some research papers claiming to present a forecast model, this article makes a point of clearly confirming that the predictor datasets considered are available going forward on a near-real-time basis, a necessity if a modeling system is actually going to be useful for real-world operational forecasting. The submission also clearly identifies (e.g., lines 400 to 410) where, when, and why each of the candidate predictor datasets are or are not useful for water supply forecasting, which is crucially important for physical credibility of the forecasts and the systems and data generating them, and which is sometimes overlooked in research around data-driven models.

Thank you for your positive and detailed feedback. We appreciate your recognition of our study's contributions and the practicality of our model for real-world forecasting. We value your recommendation for publication pending minor revisions and look forward to addressing your comments below in the revised manuscript.

That said, I think a few improvements will be needed before the paper is ready for publication. I hope the following comments will be helpful to the authors if they move forward with submitting a revised manuscript:

1. Though in general the article is well-crafted, some passages are written poorly enough that their meaning is unclear. For example, the wrong word is used, or words are used incorrectly, or elaborate vocabulary or phrasing is used when simpler wording would do.

We will carefully revise sections where wording is unclear or overly complex to improve readability and ensure the meaning is precise.

2. The literature review is not quite adequate. While there seems to be sufficient reference to prior work in the study area, this article is not just a case study, and HESS is an international journal. More broadly, the methods described here need to be placed in the wider and deeper context of previous work, not just locally but globally, in order for readers to understand its contributions and wider implications – the methods used here may be applicable in entirely different regions of the world. The wider literature does not need to be discussed in detail, nor do the methods used in this submission need to be compared against them, but the paper does need to leave some clues for readers about relevant prior publications. What stood out for me is that prior research (and practice) around seasonal water supply forecasting in western Canada and the western US, directly relevant to this study, has not been adequately acknowledged. Some points of particular note are the following (full citations are provided at the end of this review):

2.a.   It would probably be helpful to note in the article that the type of seasonal river discharge volume forecast modeling considered here is widely referred to as "water supply forecasting" (WSF) in the western North American operational hydrology and water management communities.  They don't need to use the term throughout the article, but just pointing out at the start of the paper that they're working on what is commonly called WSF will help readers connect the study to a large existing body of prior research and practice.

Thank you for this suggestion. We admit that the initial submission lacked overview of existing and /or similar practices for seasonal hydrological forecasting, especially using data-driven methods. We will expand the literature review to better place our study in a global context, particularly by referencing relevant work from North America, as you suggest, and from regions which share similar hydroclimatic and data challenges as Central Asia. We will also ensure to incorporate the cited research on water supply forecasting and teleconnection indices, ensuring our work is connected to this body of literature.

2.b.   Contrary to what seems to be implied in this article, given the way certain passages are phrased and the sparseness of literature citations, combining teleconnection indices with snow data as inputs to statistical seasonal water supply forecasting models is neither new nor rare.  It appears to have first been implemented decades ago (Garen, 1998) in the large-scale (hundreds of forecast locations) operational forecasting systems of the US Department of Agriculture's Natural Resources Conservation Service (NRCS) (Perkins et al., 2009), predictions from which are a staple for water managers across the American West.  These principal component regression models (Garen, 1992) have used a combination of SWE, accumulated precipitation, and in some cases antecedent streamflow and El Niño-Southern Oscillation indices (Garen, 1998) as predictors of seasonal river flow volumes; these methods have since be adopted widely across western North America by other operational forecast agencies.  Furthermore, continued applied R&D on combined use of snowpack observations and teleconnection indices as input variables to statistical seasonal discharge forecast models has been continued by many, such as Gobena et al (2013) in western Canada, and Moradkhani and Meier (2010) and Regonda et al. (2006a, 2006b) in the western US, to give just a few examples.  It has also been extended to discovering new climate prediction skill using nonlinear methods or new indices in areas where conventional linear teleconnections are weak, such as southern Oregon and northern California (Kennedy et al., 2009; see also Fleming and Dahlke 2014).

We appreciate your highlighting the relevant studies in 2.c. After viewing through the papers, we believe they will be valuable additions to our literature review. We will also aim to expand it with additional relevant studies.

2.c.    Though the computational model presented here appears to be novel, major elements of its philosophy and structure are strongly reminiscent of other recent advances in data-driven predictive modeling of seasonal river discharge volumes. Of note here is the multi model machine-learning metasystem (M4), which was developed for and is currently being operationally implemented by the US Department of Agriculture NRCS as its new western US-wide seasonal river discharge volume forecast model. This system has been run using in-situ SWE, precipitation, and antecedent streamflow data, as well as combinations of in-situ and remotely sensed snow data, as predictors (Fleming et al., 2021, 2024). It uses a multi-model ensemble approach in which six data-driven (statistical and machine learning) forecast systems are run independently and the results are pooled to form a best estimate, closely analogous to the modeling philosophy used in this HESS contribution. There are also significant differences between M4 and the method used in this submission, but citing M4 will better-place this HESS article's contributions in the larger research and applications literature, and provide literature support to the methods the submitted paper uses. By the same token, some related exploratory work on methods for combining outputs from multiple data-driven seasonal river discharg forecast models by Najafi and Moradkhani (2016) should be cited in this regard as well.

Thank you for these suggestions. Upon reviewing the noted models/studies, we found many concept-wise similarities and believe these will be valuable addendums to the literature review, particularly regarding methods and methodological advancements in the field.

2.d.    The foregoing are just some examples I happen to be familiar with. I'd suggest that the authors scour the literature for other prior work, including work in other regions globally, that ought to be at least briefly cited in their revised paper.

3.    The following are a few additional suggestions for improvements:

3.a.    Line 35 and elsewhere: this paper distinguishes between what it calls "dynamic" vs. "statistical" approaches. This jargon tends to be used more in other (broadly related) disciplines like regional climate modeling, with "process-based" vs. "data-driven" being more common in the operational hydrology literature. Also, it's usually "dynamical" not "dynamic", and "data-driven" also tends to be preferable to "statistical" today because of the increasing popularity of machine learning techniques (including this submission). The authors can use "dynamical" and "statistical" if they like, but to better orient readers, including the operational water resource forecasting community, to which this article seems to be in part addressed, please provide some synonyms where the terms are first introduced (line 35). It could read something like "generated using either dynamical (process-based, physics-oriented) or statistical (data-driven including machine learning and conventional statistical) modeling approaches" or something similar.

Thank you for highlighting this issue. We agree with your suggestions and will revise the manuscript to use 'process-based' instead of 'dynamic' and 'data-driven' instead of 'statistical.' We will also include synonyms for these terms where they are first introduced to better orient readers, particularly those from the operational water resource forecasting community.

> 3.b.  Line 46, "statistical forecasts of seasonal streamflow often rely solely on accumulated snowpack." Yes and no.  Yes, data on winter-spring seasonal snowpack provides the primary source of predictive skill in data-driven forecast models of spring-summer river runoff volume in snowmelt-dominated rivers.  But these models, in both the research literature and (in particular) in operational practice, at least in western North America, also almost always use additional predictor data types.  Examples include wintertime accumulated precipitation, early-season precipitation, and at some locations, antecedent streamflow and/or El Niño indices.  See point 2.b above.

Thank you for the correction, and we apologize for the confusion. What we intended to convey is that accumulated terrestrial water storage is the main determinant of seasonal water supply in snowmelt-driven basins, with snowpack being its key component. We will revise this section to avoid confusion and include a mention of other commonly used predictors.

In our study, we limited the predictors to two groups—SWE and teleconnections—because we aimed to assess the added value of teleconnections compared to SWE-based predictions. Additionally, we sought to keep the model parsimonious given the limited number of observations.

> 3.c.  Line 50: excellent point!

> 3.d.  Lines 52-53 and elsewhere: if the authors want to call the Apr-Sep target period the "vegetation season," that's fine I suppose, but it's not standard nomenclature.  Typically this would be called either the "growing season," looking at it from an agricultural water supply or broader ecological perspective, or the "runoff season", looking at it from a hydrological perspective.  And given that they call Nov-Mar the "cold season" rather than the "snowpack accumulation season", it might also be more consistent to simply call Apr-Sept the "warm season."  Overall, "growing season" seems like it might be the best fit here?

Thank you for your feedback. To ensure consistency and clarity, we will revise the manuscript to use 'growing season' for the Apr-Sep period, which is resembling the term used by local hydrometeorological agencies in Central Asia. We will also ensure that the terminology for the Nov-Mar period remains as 'cold season' for consistency with our current nomenclature.

3.e.    Figure 1: this figure is good, but for a wide international readership, please provide an additional map showing the location of the study area within the larger geographic context of Eurasia.

Thank you for your suggestion. We will include an additional map to provide geographical context for the study region.

3.f.    Line 124: predictand, not predicant

Our apologies for the mistake, we will correct this and other mistakes in the text.

3.g.    Line 125: An 18 year data record – in other words, 18 samples - is pretty short; it's enough to defensibly create one of these models, but just barely.  Commensurate limitations to the authors' ability to train model parameters and validate model predictions could be viewed as a source of uncertainty in this study; the counterargument, of course, is that with rapid climate change in mountain regions such as this study area, the statistical nonstationarity in a longer data record would have reduced its value anyway.  This might be worth a sentence or two here.  A brief explanation of why the record doesn't go back further or continue to the present could be helpful to readers as well.  My understanding of the political history of this region isn't great, but I think this was part of the Soviet Union, which (its grave misdeeds notwithstanding) wasn't too bad at keeping streamflow records, so one might have been forgiven for guessing that there might be some usable historical data here?

Thank you for highlighting this issue. Most of the data we used is derived from a previous study by Apel et al. (2018). Unfortunately, we do not have recent updates extending beyond that period until today. Moreover, available historical data spans from 1970 to the 1990s for most rivers, with more complete observations for the largest rivers (Amudarya and Naryn) up to around 2018. However, using such an extended dataset is problematic because the two datasets used to derive SWE estimates, FLDAS and GPM, are only available starting from 2000. Extending the observations further back would limit our ensemble to only ERA5 and MSWX and reduce the diversity of the ensemble, as MSWX is generated by bias-correcting ERA5 and may exhibit similar predictions for some catchments.

While we agree that overly lengthy data records can introduce non-stationarity concerns, we believe that extending the study period by including observations up to the present would have been beneficial and likely safe from such issues in our case. We acknowledge the short length of records in the current version of the discussion section and will highlight this limitation more explicitly in the introduction or data section.

3.h.    Lines 129-130: excellent point re: near-real time input data availability – this is a prerequisite for an operational forecasting model, and it's sometimes overlooked in research articles.

Thank you for your feedback.

3.i.    Lines 173-175: a little more information about the constituent models ("base models") is needed here.  What link function was used in the GLM?  And why were linear kernels used in the GP and SVR models?  Does this imply that most of the base models are essentially variants of standard, multiple linear regression?  If so, what are the pros and cons?  Note that work in the western US has shown that the relationships between winter-spring hydroclimatic forcing and spring-summer runoff response in data-driven WSF models range from nearly linear to moderately nonlinear, with clear physical explanations for these inferred functional forms (see Fleming et al., 2021).

Thank you for these guiding questions. We applied a Gaussian family link function for the GLM model. Indeed, the selected base models are linear, except for the RF model, though they differ in their approach to estimation and optimization. We experimented with several other model approaches, including using the same models with non-linear kernels. In most cases, the presented combination of models yielded similar accuracy in terms of RMSE and R-squared coefficients during LOOCV, but outperformed non-linear alternatives during testing on the hold-out sample. In some instances, depending on the basin or issue date, certain non-linear models produced slightly better predictions. However, when generalizing across all basins and issue dates, the existing structure still showed superior performance. We assume this may be due to two major and non-exclusive factors: (1) a relatively smaller number of observations and predictors, which makes non-linear machine learning models less efficient and prone to overfitting, and (2) the selection of predictors based on a primarily linear metric (Pearson's correlation).  We will reflect on these findings and their implications in the revised version of the manuscript.

3.j.    Lines 181-185: in defense of their methodological choice, which has no literature citations attached to it in the submission, the authors might wish to note that LOOCV is standard practice in western US WSF modeling; see references in point 2.b above.

Thank you for this suggestion and the references. We will appropriately refer to LOOCV as a standard practice in western US WSF modelling (Mallick et al., 2022; Granata and Di Nunno, 2024; Xu et al., 2024; Li et al., 2019).

3.k.    Lines 187-190: to improve accessibility to a broad readership which may not be uniformly well-versed in machine learning, it might be helpful to add just a sentence or two, with an additional reference or two, explaining the concept of a meta-learner.  It might also be helpful, in terms of connecting this concept to prior work in

data-driven WSF, to refer to the work of Najafi and Moradkhani (2016) on exploring different methods for creating multi-model ensembles from the predictions of several data-driven models.

Thank you for the suggestion. To improve accessibility, we will expand the description of the ensemble stacking concept in the introduction. This will be preceded by insights from the comprehensive overview by Zounemat-Kermani *et al.* (2021), which details the evolution and application of ensemble methods in hydrological prediction. We will also incorporate references to the work of Najafi and Moradkhani (2016), who explored various strategies for creating multi-model ensembles in the context of WSF.

In addition, we will cite more recent studies that demonstrate the growing application of ensemble stacking techniques in hydrological forecasting, such as: *For example, Li et al. (2019) investigated stacked ensemble models for long-term streamflow forecasting using advanced pre-processing techniques to enhance model stability. Mallick, Talukdar and Ahmed (2022) applied stacking models for real-time flood forecasting, showing significant improvement in prediction accuracy compared to individual models. Similarly, Granata and Di Nunno (2024) demonstrated the benefits of meta-learners in complex streamflow prediction tasks, while Xu et al. (2024) employed stacking methods with hybrid feature selection strategies for improved water resource management in Central Italy.*

3.l.    Line 205: in the context of operational hydrologic prediction models, data "assimilation" has a very specific connotation: formal methods for using new observational data, such as observed snowpack, to update the internal states, such as predicted snowpack, of a process-based (dynamical, physics-oriented) streamflow simulation model, often using fairly complex methods like ensemble Kalman filtering.  It is not normally used to refer to the use of some particular data type, such as snow data, as an input predictor variable in a data-driven (statistical or machine-learning) streamflow model.

Thank you for highlighting this inconsistency in the use of terminology. We will amend the terminology following your suggestions throughout the text.

3.m.    Lines 287-288: excellent point.  The authors might wish to cite literature that backs up this result, such as the excellent overview article of Hagedorn et al. (2005) and the multi-model ensemble WSF modeling article of Fleming et al. (2021).

We appreciate your positive feedback. We will place these findings in the context of the suggested references, including those previously noted on ensemble techniques in hydrology.

3.n. Figure 6: this a great illustration! I do have one question though: are all the base models used for the Vaksh and Kashkadarya rivers? It's hard to tell from the figure panels.

We appreciate your positive feedback on Figure 6. For the meta-learning model, we use base model predictions that meet a 0.2 R-squared coefficient threshold (lines 181-185). As a result, *the resulting ensembles typically consist of fewer than 16 base models (4 different models x 4 different snow inputs). We observed two trends: 1) the later the issue date, the larger the number of base models in the ensemble, and 2) larger catchments tend to include more base models, possibly due to the coarse resolution of snow products. For some rivers, such as the Vaksh and Kashkadarya, this leads to significantly fewer base models being used for ensemble stacking.* We will highlight these peculiarities in the revised version of the manuscript.

3.o. Line 348, might suggest rephrasing this in a more specific way, such as "suggest that useful near-real time SWE estimates, suitable for operational seasonal river discharge volume forecasting, can be effectively"

3.p. Line 350: "and enlarge during the snow ablation phase" – confusing wording

3.q. Lines 365-370: the entire paragraph (except for the excellent final sentence) is muddled. Please rewrite more simply and clearly.

3.r. Line 373: "confirms this assumption" – what assumption?

3.s. Lines 398, "is assumingly reasoned by their compensation" – this is meaningless, please rewrite.

3.t. Lines 400-410: excellent points.

Thank you for these suggestions. We will revise the text throughout to enhance clarity and incorporate the recommended rephrasing where applicable, ensuring that the manuscript is easily understandable.

References:

Granata, F. and Di Nunno, F.: Forecasting short- and medium-term streamflow using stacked ensemble models and different meta-learners, Stoch. Environ. Res. Risk Assess., 38, 3481–3499, https://doi.org/10.1007/s00477-024-02760-w, 2024.

Li, Y., Liang, Z., Hu, Y., Li, B., Xu, B., and Wang, D.: A multi-model integration method for monthly streamflow prediction: modified stacking ensemble strategy, J. Hydroinformatics, 22, 310–326, https://doi.org/10.2166/hydro.2019.066, 2019.

Mallick, J., Talukdar, S., and Ahmed, M.: Combining high resolution input and stacking ensemble machine learning algorithms for developing robust groundwater potentiality models in Bisha watershed, Saudi Arabia, Appl. Water Sci., 12, 77, https://doi.org/10.1007/s13201-022-01599-2, 2022.

Najafi, R. M. and Moradkhani, H.: Ensemble Combination of Seasonal Streamflow Forecasts, J. Hydrol. Eng., 21, 4015043, https://doi.org/10.1061/(ASCE)HE.1943-5584.0001250, 2016.

Xu, X., Chen, F., Wang, B., Harrison, M. T., Chen, Y., Liu, K., Zhang, C., Zhang, M., Zhang, X., Feng, P., and Hu, K.: Unleashing the power of machine learning and remote sensing for robust seasonal drought monitoring: A stacking ensemble approach, J. Hydrol., 634, 131102, https://doi.org/https://doi.org/10.1016/j.jhydrol.2024.131102, 2024.

Zounemat-Kermani, M., Batelaan, O., Fadaee, M., and Hinkelmann, R.: Ensemble machine learning paradigms in hydrology: A review, J. Hydrol., 598, 126266, https://doi.org/https://doi.org/10.1016/j.jhydrol.2021.126266, 2021.

---

## Author Comment (AC2)

Dear Reviewer,

Thank you for your detailed and valuable feedback. In response to your comments, we will refine the text to better reflect the specific contributions of our study. We acknowledge that teleconnections and the use of SWE in streamflow forecasting are well-established, and we will clarify our findings by focusing on the operational insights specific to Central Asia. The literature review will be expanded to better contextualize our contribution, and we will more comprehensively cite other relevant work, including studies from North America and other regions.

Regarding forecast uncertainty (comment 2), we will incorporate analysis bootstrapped prediction intervals and Q-Q plots to better quantify forecast uncertainty. In addition, to address concerns about the limited sample size (comment 3), we will reduce the number of predictors to a maximum of three and revise the validation approach by removing the hold-out sample. Instead of the hold-out validation sample, we will perform full-sample LOOCV, ensuring more robust evaluation. We believe that the suggested changes, along with updates to figures and more consistent terminology, will improve the clarity and rigor of the manuscript. Please find below our responses (in blue) to your referee comments (in black).

With regards,

The authors.

In this manuscript, the authors explore the relative contribution of large-scale climate oscillation predictors and snow water equivalent on the quality of April-September seasonal streamflow forecasts in eight catchments located across the Pamir and Tian-Shan mountains (central Asia). To this end, the authors first examine the correlation between climate modes of variability and (i) catchment-scale precipitation over the peak precipitation season (February-July), and (ii) April-September seasonal streamflow. Then, the authors adjust 16 models resulting from the combination of four statistical models and four SWE products, using SWE (at four forecast initialization times) as one of the predictors, and large scale climate indices as additional predictors. The total sample size (i.e., 18 points obtained from 18 years with data) is split into a sample of 15 points for cross-validation, and the remaining points are used for additional testing. The authors conclude that their technique is "a novel way to reduce uncertainties in seasonal discharge predictions in data-scarce snowmelt-dominated catchments".

This is basically a seasonal hindcasting study, generally well written and concisely presented. Nevertheless, my main critiques with this work are (1) the overselling, especially in the title, abstract and conclusions, (2) the lack of forecast uncertainty characterization (which is highlighted by the authors as a key contribution), and (3) the

limited sample size, and the way the authors address this problem in their analyses. Therefore, I think that the manuscript needs major revisions before being considered for publication in HESS.

**Major comments**

1. Title, abstract and conclusions: it is well known that the value of hydroclimatic teleconnections on seasonal streamflow forecasts is huge in snowmelt-driven catchments, especially during the preceding Fall season, when initial hydrologic conditions have not been fully developed (e.g., Mendoza *et al.*, 2017) – as the authors write in L22-24, and conclude in L403-404. There is a long history on the use of large-scale climate information for seasonal streamflow forecasting (e.g., Piechota *et al.*, 1998), and what the authors state in L20-21 and other parts of the manuscript was neatly shown nearly two decades ago using custom-based climate indices in two western US catchments (see Figure 8 in Grantz *et al.*, 2005; and also Regonda *et al.*, 2006; Opitz-Stapleton *et al.*, 2007; Bracken *et al.*, 2010; Mendoza *et al.*, 2014, etc.). Additionally, the use of simulated catchment-averaged SWE as a predictor to feed statistical models (L105-106) is not new either (e.g., Rosenberg *et al.*, 2011; Mendoza *et al.*, 2017). In other words, the findings reported by the authors are not novel and, based on this, I think that they should refine the title, abstract and conclusions to make them more specific to their actual contribution to the existing literature.

Thank you for your detailed feedback and for highlighting that teleconnections have been explored in seasonal streamflow forecasting, including for snowmelt-driven catchments. As you correctly noted, we mention this in the manuscript, although those excerpts were succinct and warrant to be expanded. Our manuscript offers additional contributions that expand on the valuable insights from Mendoza et al. (2017) and other researchers. As noted in the abstract and discussion, our study identifies specific instances when teleconnections may become more influential: at extended lead times, during strong in-season climate variability, or when catchment snow estimates are less reliable. We believe that this context, which has received little attention in previous studies (except for the first instance, which aligns with findings of Mendoza et al. (2017), helps refine the use of teleconnections for operational water supply forecasting.

Furthermore, while acknowledging that the use of large-scale climate indices and snow data for such forecasting has been previously explored, most of the cited references focus on North America. We believe our study offers novelty by demonstrating how teleconnections are pertinent to Central Asia and how their inclusion can aid seasonal water supply forecasting.

We will expand literature review on use of climate teleconnections in seasonal water supply forecasting, and clarify the abovementioned distinctions in the manuscript's title, Abstract, and Discussion to better reflect our study's contributions.

2. L25: the authors declare that their approach "provides a novel way to reduce uncertainties in seasonal discharge prediction". Do they refer to the spread of seasonal forecasts? Although they describe an ensemble stacking framework to produce a final forecast, only deterministic evaluation metrics (coefficient of determination and normalized mean absolute error) are reported, and no characterizations of hydrological prediction uncertainties are presented. A popular to do so is through ensembles (Georgakakos *et al.*, 2004; also, see publications produced by the HEPEX community on this topic), analyzing, for example, the statistical consistency of seasonal forecasts with graphical devices like rank histograms (Hamill, 2001) or Q-Q plot (Renard *et al.*, 2010), complementing with ensemble verification metrics (e.g., De Lannoy *et al.*, 2006). Therefore, I recommend the authors to take advantage of the multiple models developed to characterize forecast uncertainty or, alternatively, delete any references to "forecast uncertainty" from their manuscript (which I think would diminish the quality of their research).

Thank you for your valuable feedback. To address this well-grounded point, we propose implementing an uncertainty assessment using a bootstrapping approach. Our ensemble stacking approach involves different numbers of models per basin and issue date, which could complicate direct comparisons of ensemble spread. We suggest using bootstrapping to both resample the data and retrain the SVM meta-learner for each bootstrap sample to fully capture forecast uncertainty. In this framework, LOOCV will be used for training and assessing the generalizability of both the base models and the meta-learner. Afterwards the bootstrapping will be applied by training SVM meta-learner on each bootstrapped sample to generate 90% prediction intervals based on the variability in bootstrapped predictions. To complement this uncertainty characterization, we will also implement Q-Q plots to visually assess the consistency between predicted and observed discharge values.

3. Sample size (L126-127): this is a major issue in seasonal streamflow forecasting, since only one training/verification point is available per year. Therefore:In my opinion, the sample size is not large enough to support – being extremely generous – more than three predictor variables in their models (the authors report up to five predictors in Figure 5 for the Chu River basin), given the high risk of overfitting (see Wilks, 2011 or any other book on Statistics). Hence, I think that the authors should revisit their statistical models, removing combinations of predictors that may introduce multicollinearity.

Thank you for highlighting the issue of small sample size. Most of the data we used is derived from a previous study by Apel et al. (2018) . Unfortunately, we do not have recent updates extending beyond that period until today. Moreover, available historical data spans from 1970 to the 1990s for most rivers, with more complete observations for the largest rivers (Amudarya and Naryn) up to around 2018. However, using this extended dataset is problematic because the two datasets used to derive SWE estimates, FLDAS and GPM, are only available starting

from 2000. Extending the observations further back would limit our ensemble to only ERA5 and MSWX, reducing the diversity of the ensemble, as MSWX is generated by bias-correcting ERA5 and may exhibit similar predictions for some catchments.

We acknowledge the limitations imposed by a small sample size, which could impact the generalizability of our results. To address this, we integrate several strategies: we employ an ensemble approach which is particularly effective for addressing the challenges associated with small datasets by combining the strengths of multiple models (Dietterich, 2000; Zounemat-Kermani et al., 2021). Furthermore, we adhere to parsimony in model selection and parametrization, and therefore we employ relatively simple machine learning models with parameters fixed at conservative level to minimize overfitting. To further enhance the robustness of the framework given the limited length of observations, we incorporate multiple independent data sources into the ensemble model.

We realize that descriptions of these approaches have been succinct (e.g. lines 94-101); we will explicitly highlight these strategies in a revised version of the manuscript, linking them directly to the limitations posed by the small data sample. To address your concern and ensure a balance between model complexity and interpretability, we will also reduce the number of predictors to a maximum of three in a revised version of the manuscript.

While we acknowledge that multicollinearity can distort the interpretation of individual predictor effects, evidence suggests it is less problematic for predictive performance (Kiers and Smilde, 2007).The selected model types, especially Support Vector Machines (SVM) and Random Forests (RF), are inherently more robust to multicollinearity and can accommodate more predictor variables than observations without a loss in predictive power. A preliminary check of collinearity among the predictors revealed that Pearson's correlation coefficients are generally below 0.1, except for PDO and SOI at their selected months (used in two basins), where the coefficient reaches 0.55. While we are unsure if this constitutes strong multicollinearity, to be cautious with the interpretation of results, we propose showing variable importance (Figure 5) aggregated into two classes: SWE and climate indices.

> I do not think it is appropriate to split their sample of points (n = 18) into a smaller sample for leave-one-out cross validation (with n =15), and another sample for verification that contains three (L314) or even two points. I recommend the authors using the entire sample to perform cross-validation and compute verification metrics. Further, they should characterize the impact of sampling uncertainty, which could be done by adding confidence intervals created through bootstrapping with replacement (see section 5.5 in Araya et al., 2023). This is a critical point that the authors should address, given the very small sample size.

Thank you for your concern regarding the adequacy of the training sample and the subsequent suggestions. Our two-tiered validation approach, combining LOOCV on the training sample with hold-out validation on the testing data, was intended as additional element for checking forecast reliability. However, we acknowledge that the hold-out validation sample, consisting of only 2 to 3 observations, may appear unrepresentative. In line with your suggestions, we will extend the training sample by removing the hold-out validation and incorporating bootstrap-based prediction intervals for the predictions.

**Specific comments**

4. L13: The authors use the term "predictions", which is an excessively ample word for what they really do. In this line, I recommend the authors using the word "forecasts", and consider using the words "hindcasts" and "hindcasting" in the remainder of the manuscript, especially when describing their methods and results (please see section 3 in Beven and Young, 2013).

Thank you for your suggestion. We will replace "predictions" with "forecasts" and "hindcasts" as appropriate.

5. L30: This population estimate is for almost ten years old. I suggest updating the number and the reference.

We could not find updates to this estimate in the given context. Immerzeel et al. (2020) providea similar estimate of ~1.9 billion people, though they focus on populations dependent on mountains. If you are aware of newer estimates, we would appreciate it if you could share the relevant reference with us.

6. L35: Sometimes you use "dynamic", and sometimes "dynamical". Please pick one term and be consistent.

Thank you for highlighting this inconsistency. We will revise the manuscript to use 'process-based' instead of "dynamic/dynamical" and "data-driven" instead of "statistical".

7. L36-37: This sentence is incorrect. Climate forecasts are not used until the IHCs have been produced by running a model with a historical meteorological dataset up to the forecast initialization time.

We appreciate this comment and apologize for the confusion. We intended to convey the same point, but used incorrect wording. In the revised version of the manuscript, the sentence will read: "*Process-based forecasts use a hydrological or land-surface model to estimate current hydrologic conditions, typically with assimilation of observational data, followed by the use of climate forecasts to project future conditions.*"

8. L39: I disagree with the authors' statement, since computational demand depends on model complexity and, therefore, a model simulation might take from seconds (e.g., GR4J, SAC-SMA) to several minutes (e.g., VIC, SUMMA) in a home PC.

Thank you for highlighting this. We agree that computational demand depends on model complexity. However, depending on the type of model and the level of spatial resolution, a simulation can take significantly longer than just a few minutes. Since the paragraph compares process-based and data-driven modelling approaches for hydrological forecasting, we suggest splitting the sentence into two, with the new sentence reading as: "*Process-based models typically exhibit higher computational demands.*"

9. L40: Note that meteorological variables obtained from numerical climate models ARE prone to uncertainties.

Thank you for bringing this to our attention. We will revise the wording accordingly.

10. L45-46: I think that the authors should cite more papers when referring to the relevance of SWE as a predictor in mountainous catchments (e.g., Garen, 1992; Rosenberg et al., 2011; Mendoza et al., 2014). In general, I recommend the authors strengthening the literature review in this paragraph.

Thank you for this suggestion. We admit that the initial submission lacked overview of existing and /or similar practices for seasonal hydrological forecasting based on accumulated snowpack, especially using data-driven methods. We will expand the literature review to better place our study in a global context, particularly by referencing relevant work from North America, as you suggest, and possibly from regions which share similar hydroclimatic and data challenges as Central Asia.

11. L46: "statistical forecasts of seasonal streamflow often rely solely on accumulated snowpack". I disagree with this statement. The current operational systems managed by the NRCS for the western US and the DGA for Chile use, besides SWE, in situ measurements of precipitation, air temperature and streamflow measured in the preceding months.

Thank you for the correction; we apologize for the confusion. What we intended to convey is that accumulated terrestrial water storage is the main determinant of seasonal water supply, with snowpack being its key component. We will revise this section to avoid confusion and include a mention of other commonly used predictors. In our study, we limited the predictors to two groups—SWE and teleconnections—because we aimed to assess the added value of teleconnections compared to SWE-based predictions. Additionally, we sought to keep the model parsimonious given the limited number of observations.

12. L74: Are the authors referring to hydrological droughts? I think that any paper by Anne Van Loon (e.g., Van Loon, 2015) may be useful to clarify this point.

Thank you for the suggested references. In this sentence we are referring to seasonal precipitation levels lower than the historical norm, which may represent droughts. However as this paragraph aims to overview climate teleconnections relevant to the Central Asian region, rather than droughts, we find it challenging to refer to Van Loon (2015) in this specific context.

13. L88-89: This approach was proposed and tested more than two decades ago (e.g., Piechota et al., 1998).

14. L97: It would be good clarifying here that SWE can be directly obtained from reanalysis, or estimated by combining satellite remotely sensed snow depth and a snow density model.

15. L99-101: Please note that ensemble techniques have been used for decades in seasonal streamflow forecasting (e.g., Twedt et al., 1977; Day, 1985; Regonda et al., 2006; Wang et al., 2011; Arnal et al., 2018; Emerton et al., 2018; Lucatero et al., 2018; Girons Lopez et al., 2021; Araya et al., 2023).

Thank you for these suggestions. We will expand the literature review accordingly, with appropriate referencing to earlier studies.

16. Table 1: I suggest adding the period used to compute the variables and more hydroclimatic descriptors, like mean annual runoff (mm/yr), mean annual runoff ratio and aridity index. Please change the units of seasonal discharge to mm/yr,

Thank you for these suggestions, we will amend descriptive statistics accordingly.

17. L173: what link function did you use in your GLM?

We applied a Gaussian family link function for the GLM model.

18. L189: Looks like the SVR works as a post-processor, right?

Yes, the SVR functions as a post-processor in our ensemble stacking approach.

19. L190-191: given the small sample size, I recommend deleting this step from your workflow (see comment #3).

Thank you for the suggestion. We agree to revise our validation strategy in line with your recommendation in comment #3. All relevant sections of the manuscript will be amended accordingly.

> 20. L205, L206, L297, L351 and L353 and everywhere else: the authors use the term "assimilate" when referring to the use of modeled SWE as a predictor in their statistical model. Nevertheless, such term is typically used when referring to a family of techniques that combine imperfect models with uncertain observations to improve dynamical model estimates (e.g., Liu and Gupta, 2007; Reichle, 2008; Kumar et al., 2016; Smyth et al., 2022). Since the authors do not refer to the former concept anywhere in this manuscript, I suggest deleting the words "assimilate" or "assimilation".

Thank you for highlighting this inconsistency in used terms. We will amend the terms used accordingly throughout the text.

> 21. L221: what do you mean with the word "underperforming"?

Thank you for your comment. By "underperforming" we refer to the fact that certain SWE products exhibit lower predictive accuracy in specific catchments compared to other products. We will clarify this in the revised manuscript to ensure the sentence is clear.

> 22. L221-222: I think that this sentence contradicts the previous one. Also, if ERA5-L and MSWX are better, why don't you just pick one of these products for subsequent analyses? Some of your subsequent figures are unnecessarily complicated.

Thank you for this comment. Our intention was to convey that, in general, ERA5-L and MSWX-based estimates show higher correlations with seasonal streamflow. However, different SWE products perform better or worse depending on the catchment, which is why we have not selected a single product for subsequent analyses. Figure 3 is intended not only to illustrate the association between snowpack and seasonal streamflow and how this relationship changes across forecast issue dates, but also to highlight the differences between the snow estimates. For these reasons, we would like to retain both the figure and its explanations in the text.

> 23. Section 5.2 and Figure 4: since your target variable is seasonal streamflow, you could show correlation results between this variable and climate indices here, and move the correlation results with precipitation to supplementary material.

We appreciate this suggestion. We propose retaining the correlation graph between peak-season precipitation and climate indices in the main text, while moving the streamflow correlation graph to the supplementary material. We believe this graph provides valuable context, which we will elaborate on, regarding the associations between climate oscillations

and interannual precipitation variability, which in turn influences interannual fluctuations in streamflow levels.

24. Figure 5: I do not think you can support more than three predictors with a sample size n = 18 (see comment #3).

The new version of Figure 5 will display only three predictors, as noted in our response to the comment #3.

25. L295: Do you mean winner among statistical models? Can you please be more specific?

Thank you for your comment. To clarify, by 'best-performing,' we meant the model that most accurately predicts streamflow across all catchments and forecast lead times. We will revise the manuscript to make this clear.

26. L301-302: I do not think that the authors are quantifying uncertainty (see comment #2).

27. Figure 6 is quite difficult to read. Since the focus of the paper is on the relevance of climate information in seasonal streamflow forecasting, why don't you just show the best-performing statistical model, with the best SWE product? Further, you should include the assessment period in each figure caption.

Thank you for your suggestion. We would like to retain Figure 6, as it not only displays the accuracy of both the base model forecasts and the final ensemble forecast, but it also illustrates the varying performance of the base models across different issue dates. It also conveys the message that the ensemble forecast outperforms single model forecasts. We believe this broader comparison is useful for demonstrating the added value of the ensemble approach. Regarding the assessment periods, since they will be indicated in a previous figure/table (see our response to the comment #16), we do not see the need to repeat them again in this figure. However, we will ensure the figure captions are clear and include all necessary information.

28. L313: This is not true for all catchments. See, for example, the red bars for the Kashkadarya and Chu basins.

Thank you for highlighting this. Since the revised version will now include all observations in the LOOCV with no hold-out validation sample, this and related text exerts will likely be removed from the text.

29. L322: Do you mean larger errors? Are you comparing against the results obtained with SWE and climate information? In that case, I really think you should define a Skill Score for a comparative assessment.

Thank you for the comment. Yes, we are comparing two configurations of the same models—one using only SWE and the other using both SWE and climate indices. To ensure a more consistent comparison, and in light of previous suggestions (removing hold-out validation), we propose including a single graph that displays the MAEs of both configurations for each basin and issue date. We will also consider, as an alternative, displaying only the incremental differences between the two configurations as the percentage reduction in MAE for each basin and issue date.

30. Figure 8: I recommend presenting these results using scatter plots (eight panels), along with the 1:1 line, percent bias, MAE and $R^2$.

We appreciate this suggestion. We will display these results as Q-Q plots with embedded percent bias, MAE and $R^2$.

31. L348: What do you mean with 'effectively'? That near real-time SWE estimates are actually useful for seasonal streamflow forecasting?

By 'effectively,' we intended to convey that SWE estimates derived from or modelled using global sources, despite their biases and spatio-temporal inconsistencies, can still provide added value for seasonal streamflow forecasting. While this point may seem trivial, we believe it is relevant in the context of forecasting without in-situ data on predictors. We will consider revising this sentence to ensure clarity.

32. L350: I do not think the authors have presented any uncertainty or error propagation analysis (please see comment #2)

This sentence will be revised and updated in accordance with uncertainty analysis we proposed above (our response to the comment #2).

33. L352: Did you actually assess the accuracy of SWE products using in-situ observations?

No, as we note in the Introduction (L94-101) systematic in-situ SWE measurements are absent in the region

34. L420: In my opinion, models adjusted with such a small sample cannot be regarded as "reliable".

***Suggested edits***

35. L28: "where it sustains" -> "sustaining".

36. L32: "Accurate water availability forecasts" -> "accurate water supply forecasts".

37. L36: "current hydrologic conditions" -> "initial hydrologic conditions".

38. L42: "multiple variables" -> "multiple predictor variables".

39. L43: delete "the context of".

40. L61 and L63: replace "from now on" by "hereafter".

41. L67: delete "from satellite".

42. L73: "ENSO in its cold phase" -> "the cold phase of ENSO".

43. L74: delete "ENSO's".

44. L95-96: "used to conduct" -> "conducted".

45. L124: I think that the right word is "predictand".

46. L130: delete "in near real-time".

47. L132: "we simulated" -> "we obtained".

48. L155-156: "precipitation levels" -> "precipitation amounts".

49. L174: add "SVR" after "support vector regression".

50. L214-215:  I suggest deleting this sentence.

Thank you for the proposed edits. We will update the wording in line with your suggestions.

References:

Apel, H., Abdykerimova, Z., Agalhanova, M., Baimaganbetov, A., Gavrilenko, N., Gerlitz, L., Kalashnikova, O., Unger-Shayesteh, K., Vorogushyn, S., and Gafurov, A.: Statistical forecast of seasonal discharge in Central Asia using observational records: development of a generic linear modelling tool for operational water resource management, Hydrol. Earth Syst. Sci., 22, 2225–2254, https://doi.org/10.5194/hess-22-2225-2018, 2018.

Dietterich, T. G.: Ensemble Methods in Machine Learning, in: Multiple Classifier Systems, 1–15,

2000.

Immerzeel, W. W., Lutz, A. F., Andrade, M., Bahl, A., Biemans, H., Bolch, T., Hyde, S., Brumby, S., Davies, B. J., Elmore, A. C., Emmer, A., Feng, M., Fernández, A., Haritashya, U., Kargel, J. S., Koppes, M., Kraaijenbrink, P. D. A., Kulkarni, A. V., Mayewski, P. A., Nepal, S., Pacheco, P., Painter, T. H., Pellicciotti, F., Rajaram, H., Rupper, S., Sinisalo, A., Shrestha, A. B., Viviroli, D., Wada, Y., Xiao, C., Yao, T., and Baillie, J. E. M.: Importance and vulnerability of the world's water towers, Nature, 577, 364–369, https://doi.org/10.1038/s41586-019-1822-y, 2020.

Kiers, H. A. L. and Smilde, A. K.: A comparison of various methods for multivariate regression with highly collinear variables, Stat. Methods Appl., 16, 193–228, https://doi.org/10.1007/s10260-006-0025-5, 2007.

Van Loon, A. F.: Hydrological drought explained, Wiley Interdiscip. Rev. Water, 2, 359–392, https://doi.org/10.1002/wat2.1085, 2015.

Mendoza, P. A., Wood, A. W., Clark, E., Rothwell, E., Clark, M. P., Nijssen, B., Brekke, L. D., and Arnold, J. R.: An intercomparison of approaches for improving operational seasonal streamflow forecasts, Hydrol. Earth Syst. Sci., 21, 3915–3935, https://doi.org/10.5194/hess-21-3915-2017, 2017.

Zounemat-Kermani, M., Batelaan, O., Fadaee, M., and Hinkelmann, R.: Ensemble machine learning paradigms in hydrology: A review, J. Hydrol., 598, 126266, https://doi.org/https://doi.org/10.1016/j.jhydrol.2021.126266, 2021.

---

## Author Response (AR1)

**RC1: Anonymous Referee #1**

Dear Reviewer,

Thank you for your valuable and constructive comments on our manuscript. Following your suggestions, we expanded the literature review to provide a broader context by referencing relevant work from North America and other regions, especially focusing on studies that utilize snowpack and climate indices for seasonal streamflow forecasting. We also refined the terminology to clarify distinctions between process-based and data-driven approaches and ensure the manuscript uses consistent, field-appropriate language. Additionally, we amended the methodological descriptions to better explain the rationale behind the chosen models and improve the clarity and precision of our wording throughout the paper. We believe that the revised version is now better aligned with current research in the field. Please find below your referee comments (in black) and our responses (in blue).

With regards,

The authors.

Review of 'The value of hydroclimatic teleconnections for snow-based seasonal streamflow forecasting,' Umirbekov et al., HESS Discussions

This submission summarizes the motivation for, and development, implementation, and performance of, a data-driven model for seasonal river runoff volume forecasting in Central Asia. The method uses as predictors a combination of SWE data from existing large-scale operational remote sensing and land surface modeling products, and indices of various atmosphere-ocean circulation patterns, as predictors; it employs a multi-model ensemble model structure, which has some machine learning elements; and is intended to serve as an actual operational forecasting tool for directly supporting water management decision-making.

My overall recommendation is for publication pending minor revisions.

The article will be an excellent contribution to HESS. The paper is succinct and well-organized. The study location considered has been historically understudied, though admittedly not so severely as some other regions of the developing world. The candidate predictor data combinations are geophysically sensible, but also to some degree original in the particular way they are used here. Though it is possible to quibble with certain aspects of the predictor selection process used, it is a reasonable and defensible approach for the task at hand. Its multi-model ensemble philosophy is fully consistent with a large body of evidence demonstrating its value, yet it is only one of a tiny handful of hydrologic modeling examples where several separate data-driven/statistical/machine learning modeling systems are used and their results pooled to form a best estimate, and it is also novel as implemented here. Unlike some research papers claiming to present a forecast model, this article makes a point of clearly confirming that the predictor datasets considered are available going forward on a near-real-time basis, a necessity if a modeling system is actually going to be useful for real-world operational forecasting. The submission also clearly identifies (e.g., lines 400 to 410) where, when, and why each of the candidate predictor datasets are or are not useful for water supply forecasting, which is crucially important for physical credibility of the forecasts and the systems and data generating them, and which is sometimes overlooked in research around data-driven models.

Thank you for your positive and detailed feedback. We appreciate your feedback on study contributions and your recommendation for publication pending minor revisions.

That said, I think a few improvements will be needed before the paper is ready for publication. I hope the following comments will be helpful to the authors if they move forward with submitting a revised manuscript:

1. Though in general the article is well-crafted, some passages are written poorly enough that their meaning is unclear. For example, the wrong word is used, or words are used incorrectly, or elaborate vocabulary or phrasing is used when simpler wording would do.

2. The literature review is not quite adequate. While there seems to be sufficient reference to prior work in the study area, this article is not just a case study, and HESS is an international journal. More broadly, the methods described here need to be placed in the wider and deeper context of previous work, not just locally but globally, in order for readers to understand its contributions and wider implications – the methods used here may be applicable in entirely different regions of the world. The wider literature does not need to be discussed in detail, nor do the methods used in this submission need to be compared against them, but the paper does need to leave some clues for readers about relevant prior publications. What stood out for me is that prior research (and practice) around seasonal water supply forecasting in western Canada and the western US, directly relevant to this study, has not been adequately acknowledged. Some points of particular note are the following (full citations are provided at the end of this review):

2.a. It would probably be helpful to note in the article that the type of seasonal river discharge volume forecast modeling considered here is widely referred to as "water supply forecasting" (WSF) in the western North American operational hydrology and water management communities. They don't need to use the term throughout the article, but just pointing out at the start of the paper that they're working on what is commonly called WSF will help readers connect the study to a large existing body of prior research and practice.

2.b. Contrary to what seems to be implied in this article, given the way certain passages are phrased and the sparseness of literature citations, combining teleconnection indices with snow data as inputs to statistical seasonal water supply forecasting models is neither new nor rare. It appears to have first been implemented decades ago (Garen, 1998) in the large-scale (hundreds of forecast locations) operational forecasting systems of the US Department of Agriculture's Natural Resources Conservation Service (NRCS) (Perkins et al., 2009), predictions from which are a staple for water managers across the American West. These principal component regression models (Garen, 1992) have used a combination of SWE, accumulated precipitation, and in some cases antecedent streamflow and El Niño-Southern Oscillation indices (Garen, 1998) as predictors of seasonal river flow volumes; these methods have since be adopted widely across western North America by other operational forecast agencies. Furthermore, continued applied R&D on combined use of snowpack observations and teleconnection indices as input variables to statistical seasonal discharge forecast models has been continued by many, such as Gobena et al (2013) in western Canada, and Moradkhani and Meier (2010) and Regonda et al. (2006a, 2006b) in the western US, to give just a few examples. It has also been extended to discovering new climate prediction skill using nonlinear methods or new indices in areas where conventional linear teleconnections are weak, such as southern Oregon and northern California (Kennedy et al., 2009; see also Fleming and Dahlke 2014).

Thank you for these suggestions. We admit that the initial submission lacked overview of similar practices for seasonal hydrological forecasting, especially using data-driven methods. We also appreciate your highlighting the relevant studies in 2.c. We expanded the Introduction with additional paragraphs (lines 51-76 in the tracked changes version of the manuscript) that provide a better context of seasonal forecasting in snowmelt dominated regions, particularly those that employ teleconnections. Relevant research mostly comes from U.S., though we also managed to locate a few similar studies in South Asia and South America. The new exerts also make a reference to term "water supply forecasting" adopted in the western U.S.

2.c. Though the computational model presented here appears to be novel, major elements of its philosophy and structure are strongly reminiscent of other recent advances in data-driven predictive modeling of seasonal river discharge volumes. Of note here is the multi model machine-learning metasystem (M4), which was developed for and is currently being operationally implemented by the US Department of Agriculture NRCS as its new western US-wide seasonal river discharge volume forecast model. This system has been run using in-situ SWE, precipitation, and antecedent streamflow data, as well as combinations of in-situ and remotely sensed snow data, as predictors (Fleming et al., 2021, 2024). It uses a multi-model ensemble approach in

which six data-driven (statistical and machine learning) forecast systems are run independently and the results are pooled to form a best estimate, closely analogous to the modeling philosophy used in this HESS contribution. There are also significant differences between M4 and the method used in this submission, but citing M4 will better-place this HESS article's contributions in the larger research and applications literature, and provide literature support to the methods the submitted paper uses. By the same token, some related exploratory work on methods for combining outputs from multiple data-driven seasonal river discharg forecast models by Najafi and Moradkhani (2016) should be cited in this regard as well.

Thank you for these suggestions. Upon reviewing the noted models/studies, we found many concept-wise similarities and believe these are valuable addendums to the literature review, particularly regarding methods and methodological advancements in the field. We have a new paragraph (lines 78-92 in the tracked changes version) which provide a brief overall overview of the methodological developments in the field.

2.d.    The foregoing are just some examples I happen to be familiar with. I'd suggest that the authors scour the literature for other prior work, including work in other regions globally, that ought to be at least briefly cited in their revised paper.

3.    The following are a few additional suggestions for improvements:

3.a.    Line 35 and elsewhere: this paper distinguishes between what it calls "dynamic" vs. "statistical" approaches. This jargon tends to be used more in other (broadly related) disciplines like regional climate modeling, with "process-based" vs. "data-driven" being more common in the operational hydrology literature. Also, it's usually "dynamical" not "dynamic", and "data-driven" also tends to be preferable to "statistical" today because of the increasing popularity of machine learning techniques (including this submission). The authors can use "dynamical" and "statistical" if they like, but to better orient readers, including the operational water resource forecasting community, to which this article seems to be in part addressed, please provide some synonyms where the terms are first introduced (line 35). It could read something like "generated using either dynamical (process-based, physics-oriented) or statistical (data-driven including machine learning and conventional statistical) modeling approaches" or something similar.

Thank you for highlighting this issue. We agree with your suggestions and revised the manuscript to use 'process-based' instead of 'dynamic' and 'data-driven' instead of 'statistical.'

3.b.    Line 46, "statistical forecasts of seasonal streamflow often rely solely on accumulated snowpack." Yes and no. Yes, data on winter-spring seasonal snowpack provides the primary source of predictive skill in data-driven forecast models of spring-summer river runoff volume in snowmelt-dominated rivers. But these models, in both the research literature and (in particular) in operational practice, at least in western North America, also almost always use additional predictor data types. Examples include wintertime accumulated precipitation, early-season precipitation, and at some locations, antecedent streamflow and/or El Niño indices. See point 2.b above.

Thank you for the correction, and we apologize for the confusion. What we intended to convey is that accumulated terrestrial water storage is the main determinant of seasonal water supply in snowmelt-driven basins, with snowpack being its key component. We amended this sentence to avoid confusion (lines 51-53). In addition, we now note this in Data section (lines 184-186). In our study, we limited the predictors to two groups—SWE and teleconnections—because we aimed to assess the added value of teleconnections compared to SWE-based predictions. Additionally, we sought to keep the model parsimonious given the limited number of observations.

3.c.    Line 50: excellent point!

3.d.    Lines 52-53 and elsewhere: if the authors want to call the Apr-Sep target period the "vegetation season," that's fine I suppose, but it's not standard nomenclature. Typically this would be called either the "growing season," looking at it from an agricultural water supply or broader ecological perspective, or the "runoff season", looking at it from a hydrological perspective. And given that they call Nov-Mar the "cold

season" rather than the "snowpack accumulation season", it might also be more consistent to simply call Apr-Sept the "warm season." Overall, "growing season" seems like it might be the best fit here?

Thank you for your feedback. To ensure consistency and clarity, the revised manuscript now uses 'growing season', which also resembles the term used by local hydrometeorological agencies in Central Asia. We also ensured that Nov-Mar period remains as 'cold season' for consistency with our current nomenclature.

3.e.    Figure 1: this figure is good, but for a wide international readership, please provide an additional map showing the location of the study area within the larger geographic context of Eurasia.

Thank you for your suggestion. Figure 1 now includes an additional map to provide geographical context for the study region.

3.f.    Line 124: predictand, not predicant

Our apologies for the mistake, we corrected the wording.

3.g.    Line 125: An 18 year data record – in other words, 18 samples - is pretty short; it's enough to defensibly create one of these models, but just barely. Commensurate limitations to the authors' ability to train model parameters and validate model predictions could be viewed as a source of uncertainty in this study; the counterargument, of course, is that with rapid climate change in mountain regions such as this study area, the statistical nonstationarity in a longer data record would have reduced its value anyway. This might be worth a sentence or two here. A brief explanation of why the record doesn't go back further or continue to the present could be helpful to readers as well. My understanding of the political history of this region isn't great, but I think this was part of the Soviet Union, which (its grave misdeeds notwithstanding) wasn't too bad at keeping streamflow records, so one might have been forgiven for guessing that there might be some usable historical data here?

Thank you for highlighting this issue. Most of the data we used is derived from a previous study by Apel et al. (2018). Unfortunately, we do not have recent updates extending beyond that period until today. Moreover, *historical data for most catchments also spans from the 1970s to the 1990s, with relatively more complete records available for the largest rivers extending up to 2000. However, incorporating these earlier records is challenging, as the datasets used to derive some SWE estimates (FLDAS and GPM) are only available from 2000 onward. Extending the observations back in time would restrict the ensemble to only ERA5-L and MSWX datasets, thereby reducing its diversity*, as MSWX is generated by bias-correcting ERA5-L and may exhibit similar predictions for some catchments. Furthermore, contrary to our earlier feedback on this point, we now agree that using records from 1970s to 1990s may introduce issues of non-stationarity. We acknowledge the short length of records in the current version of the discussion section and now highlighted this limitation more explicitly in new exert in the Data section (lines 209-219).

3.h.    Lines 129-130: excellent point re: near-real time input data availability – this is a prerequisite for an operational forecasting model, and it's sometimes overlooked in research articles.

Thank you for your feedback.

3.i.    Lines 173-175: a little more information about the constituent models ("base models") is needed here. What link function was used in the GLM? And why were linear kernels used in the GP and SVR models? Does this imply that most of the base models are essentially variants of standard, multiple linear regression? If so, what are the pros and cons? Note that work in the western US has shown that the relationships between winter-spring hydroclimatic forcing and spring-summer runoff response in data-driven WSF models range from nearly linear to moderately nonlinear, with clear physical explanations for these inferred functional forms (see Fleming et al., 2021).

Thank you for these guiding questions. We applied a Gaussian family link function for the GLM model. Indeed, the selected base models are linear, except for the RF model, though they differ in their approach to estimation and optimization. We tested several other ML techniques as base models, including using the same models with non-linear kernels. In most cases, the presented combination of models yielded a better accuracy in terms of MAE and R-squared coefficients during LOOCV. In some instances, depending on the basin or issue date, certain non-linear models produced slightly better predictions. However, when generalizing across all basins and issue dates, the existing structure still showed superior performance. We assume this may be due to two major and non-exclusive factors: (1) a relatively smaller number of observations and predictors, which makes non-linear machine learning models less efficient and prone to overfitting, and (2) the selection of predictors based on a linear metric (Pearson's correlation) may have inherently favored linear models. We have now reflected on these findings in lines 410-418 of the revised version of the manuscript.

3.j. Lines 181-185: in defense of their methodological choice, which has no literature citations attached to it in the submission, the authors might wish to note that LOOCV is standard practice in western US WSF modeling; see references in point 2.b above.

Thank you for this suggestion and the references. Now, we now also refer to LOOCV as a standard practice in western US WSF modelling with corresponding references in lines 267-268.

3.k. Lines 187-190: to improve accessibility to a broad readership which may not be uniformly well-versed in machine learning, it might be helpful to add just a sentence or two, with an additional reference or two, explaining the concept of a meta-learner. It might also be helpful, in terms of connecting this concept to prior work in data-driven WSF, to refer to the work of Najafi and Moradkhani (2016) on exploring different methods for creating multi-model ensembles from the predictions of several data-driven models.

Thank you for the suggestion. To improve accessibility, we included a brief description of the ensemble stacking concept, and provided some references on the respective applications in hydrological sciences in lines 242-246.

3.l. Line 205: in the context of operational hydrologic prediction models, data "assimilation" has a very specific connotation: formal methods for using new observational data, such as observed snowpack, to update the internal states, such as predicted snowpack, of a process-based (dynamical, physics-oriented) streamflow simulation model, often using fairly complex methods like ensemble Kalman filtering. It is not normally used to refer to the use of some particular data type, such as snow data, as an input predictor variable in a data-driven (statistical or machine-learning) streamflow model.

Thank you for highlighting this inconsistency in the use of terminology. We amended the wording throughout the text following your suggestions .

3.m. Lines 287-288: excellent point. The authors might wish to cite literature that backs up this result, such as the excellent overview article of Hagedorn et al. (2005) and the multi-model ensemble WSF modeling article of Fleming et al. (2021).

We appreciate your positive feedback. We have supplemented the paragraph with a respective sentence in lines 381-383.

3.n. Figure 6: this a great illustration! I do have one question though: are all the base models used for the Vaksh and Kashkadarya rivers? It's hard to tell from the figure panels.

We appreciate your positive feedback on Figure 6. For the meta-learning model, we used base model predictions that meet a 0.2 R-squared coefficient threshold (noted in lines 268-271) In addition, we now also note (lines 385-389) ".. *the resulting stacked ensembles typically consist of fewer than 16 base models. We observe two trends in this regard: (1) the later the issue date, the greater the number of base models included in the ensemble, and (2) larger catchments tend to incorporate more base models.*"

3.o.   Line 348, might suggest rephrasing this in a more specific way, such as "suggest that useful near-real time SWE estimates, suitable for operational seasonal river discharge volume forecasting, can be effectively"

3.p.   Line 350: "and enlarge during the snow ablation phase" – confusing wording

3.q.   Lines 365-370: the entire paragraph (except for the excellent final sentence) is muddled.  Please rewrite more simply and clearly.

3.r.   Line 373: "confirms this assumption" – what assumption?

3.s.   Lines 398, "is assumingly reasoned by their compensation" – this is meaningless, please rewrite.

3.t.   Lines 400-410: excellent points.

Thank you for these suggestions. We revised the text to enhance clarity.

**RC2: Anonymous Referee #2**

Dear Reviewer,

Thank you for your detailed and valuable feedback. In response to your comments, we refined the text to better reflect the specific contributions of our study. We acknowledge that teleconnections and the use of SWE in streamflow forecasting are well-established, and we clarify our findings by focusing on the operational insights specific to Central Asia. The literature review is now expanded to better contextualize our contribution, and we refer to other relevant work, including studies from North America and other regions.

Regarding forecast uncertainty (comment 2), we incorporated analysis of bootstrap-based prediction uncertainty of the forecasts and Q-Q plots. In addition, to address concerns about the limited sample size (comment 3), we reduced the number of predictors to a maximum of three and did not split the data to LOOCV and hold-out subsamples. We believe that the suggested changes, along with updates to figures and more consistent terminology, improved the clarity and rigor of the manuscript. Please find below our responses (in blue) to your referee comments (in black).

With regards,

The authors.

In this manuscript, the authors explore the relative contribution of large-scale climate oscillation predictors and snow water equivalent on the quality of April-September seasonal streamflow forecasts in eight catchments located across the Pamir and Tian-Shan mountains (central Asia). To this end, the authors first examine the correlation between climate modes of variability and (i) catchment-scale precipitation over the peak precipitation season (February-July), and (ii) April-September seasonal streamflow. Then, the authors adjust 16 models resulting from the combination of four statistical models and four SWE products, using SWE (at four forecast initialization times) as one of the predictors, and large-scale climate indices as additional predictors. The total sample size (i.e., 18 points obtained from 18 years with data) is split into a sample of 15 points for cross-validation, and the remaining points are used for additional testing. The authors conclude that their technique is "a novel way to reduce uncertainties in seasonal discharge predictions in data-scarce snowmelt-dominated catchments".

This is basically a seasonal hindcasting study, generally well written and concisely presented. Nevertheless, my main critiques with this work are (1) the overselling, especially in the title, abstract and conclusions, (2) the lack of forecast uncertainty characterization (which is highlighted by the authors as a key contribution), and (3) the limited sample size, and the way the authors address this problem in their analyses. Therefore, I think that the manuscript needs major revisions before being considered for publication in HESS.

*Major comments*

1. Title, abstract and conclusions: it is well known that the value of hydroclimatic teleconnections on seasonal streamflow forecasts is huge in snowmelt-driven catchments, especially during the preceding Fall season, when initial hydrologic conditions have not been fully developed (e.g., Mendoza *et al.*, 2017) – as the authors write in L22-24, and conclude in L403-404. There is a long history on the use of large-scale climate information for seasonal streamflow forecasting (e.g., Piechota *et al.*, 1998), and what the authors state in L20-21 and other parts of the manuscript was neatly shown nearly two decades ago using custom-based climate indices in two western US catchments (see Figure 8 in Grantz *et al.*, 2005; and also Regonda *et al.*, 2006; Opitz-Stapleton *et al.*, 2007; Bracken *et al.*, 2010; Mendoza *et al.*, 2014, etc.). Additionally, the use of simulated catchment-averaged SWE as a predictor to feed statistical models (L105-106) is not new either (e.g., Rosenberg *et al.*, 2011; Mendoza *et al.*, 2017). In other words, the findings reported by the authors are not novel and, based on this, I think that they should refine the title, abstract and conclusions to make them more specific to their actual contribution to the existing literature.

Thank you for your detailed feedback and for highlighting that teleconnections have been explored in seasonal streamflow forecasting, including for snowmelt-driven catchments. As you correctly noted, we mention this in the manuscript, although those excerpts were succinct and warrant to be expanded. Our manuscript offers additional contributions that expand on the valuable insights from Mendoza et al. (2017) and other researchers. As noted in the abstract and discussion, our study identifies specific instances when teleconnections may become more influential: at extended lead times, during strong in-season climate variability, or when catchment snow estimates are less reliable. Furthermore, while acknowledging that the use of large-scale climate indices and snow data for such forecasting has been previously explored, most of the cited references focus on North America. We believe our study offers novelty by demonstrating how teleconnections are pertinent to Central Asia and how their inclusion can aid seasonal water supply forecasting.

We expanded literature review on use of climate teleconnections in seasonal water supply forecasting (lines 51-59 in the tracked changes version of the manuscript), and amended the manuscript's Title, Abstract, and Discussion to better reflect our study's contributions.

> 2. L25: the authors declare that their approach "provides a novel way to reduce uncertainties in seasonal discharge prediction". Do they refer to the spread of seasonal forecasts? Although they describe an ensemble stacking framework to produce a final forecast, only deterministic evaluation metrics (coefficient of determination and normalized mean absolute error) are reported, and no characterizations of hydrological prediction uncertainties are presented. A popular to do so is through ensembles (Georgakakos *et al.*, 2004; also, see publications produced by the HEPEX community on this topic), analyzing, for example, the statistical consistency of seasonal forecasts with graphical devices like rank histograms (Hamill, 2001) or Q-Q plot (Renard *et al.*, 2010), complementing with ensemble verification metrics (e.g., De Lannoy *et al.*, 2006). Therefore, I recommend the authors to take advantage of the multiple models developed to characterize forecast uncertainty or, alternatively, delete any references to "forecast uncertainty" from their manuscript (which I think would diminish the quality of their research).

Thank you for your valuable feedback. To address this well-grounded point, we integrated uncertainty assessment with bootstrapping by resampling the data and retraining the SVM meta-learner on each bootstrapped sample. In this way we estimated 80% prediction intervals based on the variability across bootstrapped forecasts. To complement this uncertainty characterization, we also implemented Q-Q plots to assess the consistency between predicted and observed discharge values. These results are incorporated into the manuscript as a new subsection 5.4 Predictive uncertainty (lines 459 -496 in the tracked changes version)

> 3. Sample size (L126-127): this is a major issue in seasonal streamflow forecasting, since only one training/verification point is available per year. Therefore: In my opinion, the sample size is not large enough to support – being extremely generous – more than three predictor variables in their models (the authors report up to five predictors in Figure 5 for the Chu River basin), given the high risk of overfitting (see Wilks, 2011 or any other book on Statistics). Hence, I think that the authors should revisit their statistical models, removing combinations of predictors that may introduce multicollinearity.

Thank you for highlighting the issue of small sample size. Most of the data we used is derived from a previous study by Apel et al. (2018) . Unfortunately, we do not have recent updates extending beyond that period until today. Moreover, available historical data spans from 1970 to the 1990s for most rivers, with more complete observations for the largest rivers (Amudarya and Naryn) up to around 2018. However, using this extended dataset is problematic because the two datasets used to derive SWE estimates, FLDAS and GPM, are only available starting from 2000. Extending the observations further back in time would limit our ensemble to only ERA5-L and MSWX, reducing the diversity of the ensemble, as MSWX is generated by bias-correcting ERA5-L and may exhibit similar predictions for some catchments.

We acknowledge the limitations imposed by a small sample size, which could impact the generalizability of our results. To address this, we integrated several strategies: we employed an ensemble approach which is particularly effective for addressing the challenges associated with small datasets by combining the strengths of multiple models (Dietterich, 2000; Zounemat-Kermani et al., 2021). Furthermore, we adhered to parsimony in model selection and parametrization, and therefore we employed relatively simple machine learning models with parameters fixed at

conservative level to minimize overfitting. To further enhance the robustness of the framework given the limited length of observations, we incorporated multiple independent data sources into the ensemble model.

Since descriptions of these approaches have been succinct in the initial version of the manuscript, we had explicitly highlighted these strategies in a revised version of the manuscript, linking them directly to the limitations posed by the small data sample (e.g. lines 89-92, 143-145, 209-219). We also reduced the number of predictors to a maximum of three in a revised version of the manuscript (noted in lines 251-252).

While we acknowledge that multicollinearity can distort the interpretation of individual predictor effects, evidence suggests it is less problematic for predictive performance (Kiers and Smilde, 2007). The selected model types, especially Support Vector Machines (SVM) and Random Forests (RF), are inherently more robust to multicollinearity and can accommodate more predictor variables than observations without a loss in predictive power. A brief check of collinearity among the predictors revealed that correlation coefficients are generally low, except for PDO and SOI at their selected months (used in two basins), where the coefficient reaches 0.55. While we are unsure if this constitutes strong multicollinearity, to be cautious with the interpretation of results, we have amended a set of predictors per each basin so that they either include SOI or PDO (Figure 5). For reference, we also included Figure S2 in the Supplement, which shows a density histogram of all pairwise correlations of the predictors across all basins and issue dates.

> I do not think it is appropriate to split their sample of points (n = 18) into a smaller sample for leave-one-out cross validation (with n =15), and another sample for verification that contains three (L314) or even two points. I recommend the authors using the entire sample to perform cross-validation and compute verification metrics. Further, they should characterize the impact of sampling uncertainty, which could be done by adding confidence intervals created through bootstrapping with replacement (see section 5.5 in Araya et al., 2023). This is a critical point that the authors should address, given the very small sample size.

Thank you for your concern regarding the adequacy of the training sample and the subsequent suggestions. Our two-tiered validation approach, combining LOOCV on the training sample with hold-out validation on the testing data, was intended as additional element for checking forecast reliability. However, we acknowledge that the hold-out validation sample, consisting of only 2 to 3 observations, may appear unrepresentative. In line with your suggestions, we removed the hold-out validation and incorporated bootstrap-based prediction intervals.

**Specific comments**

> 4. L13: The authors use the term "predictions", which is an excessively ample word for what they really do. In this line, I recommend the authors using the word "forecasts", and consider using the words "hindcasts" and "hindcasting" in the remainder of the manuscript, especially when describing their methods and results (please see section 3 in Beven and Young, 2013).

Thank you for your suggestion. We replaced "predictions" with "forecasts" and "hindcasts" as appropriate.

> 5. L30: This population estimate is for almost ten years old. I suggest updating the number and the reference.

We could not find updates to this estimate in the given context. Immerzeel et al. (2020) provide a similar estimate of ~1.9 billion people, though they focus on populations dependent on mountains.

> 6. L35: Sometimes you use "dynamic", and sometimes "dynamical". Please pick one term and be consistent.

Thank you for highlighting this inconsistency. We changed the wording throughout the manuscript to use 'process-based' instead of "dynamic/dynamical" and "data-driven" instead of "statistical".

7. L36-37: This sentence is incorrect. Climate forecasts are not used until the IHCs have been produced by running a model with a historical meteorological dataset up to the forecast initialization time.

*We appreciate this comment and apologize for the confusion. We intended to convey the same point, but used incorrect wording. In the revised version of the manuscript, the sentence reads as: "Process-based forecasts use a hydrological or land-surface model to estimate current hydrologic conditions, typically with assimilation of observational data, followed by the use of climate forecasts to project future conditions." (lines 39-41)*

8. L39: I disagree with the authors' statement, since computational demand depends on model complexity and, therefore, a model simulation might take from seconds (e.g., GR4J, SAC-SMA) to several minutes (e.g., VIC, SUMMA) in a home PC.

*Thank you for highlighting this. We agree that computational demand depends on model complexity. However, depending on the type of model and the level of spatial resolution, a simulation can take significantly longer than just a few minutes. Since the paragraph compares process-based and data-driven modelling approaches for hydrological forecasting, we split the original sentence into two, with the new sentence: "In addition, process-based models typically exhibit higher computational demands." (lines 46-47)*

9. L40: Note that meteorological variables obtained from numerical climate models ARE prone to uncertainties.

*Thank you for bringing this to our attention. We revised the wording accordingly.*

10. L45-46: I think that the authors should cite more papers when referring to the relevance of SWE as a predictor in mountainous catchments (e.g., Garen, 1992; Rosenberg et al., 2011; Mendoza et al., 2014). In general, I recommend the authors strengthening the literature review in this paragraph.

*Thank you for this suggestion. We admit that the initial submission lacked overview of existing practices for seasonal hydrological forecasting based on accumulated snowpack, especially using data-driven methods. We expanded the literature review to better place our study in a global context, particularly by referencing relevant work from North America, but also including similar studies in other regions (lines 51-76).*

11. L46: "statistical forecasts of seasonal streamflow often rely solely on accumulated snowpack". I disagree with this statement. The current operational systems managed by the NRCS for the western US and the DGA for Chile use, besides SWE, in situ measurements of precipitation, air temperature and streamflow measured in the preceding months.

*Thank you for the correction; we apologize for the confusion. What we intended to convey is that accumulated terrestrial water storage is the main determinant of seasonal water supply, with snowpack being its key component. We revised this sentence to avoid confusion (lines 51-52). In addition, the Data section has a new exert which notes additional predictors (lines 184-188).*

12. L74: Are the authors referring to hydrological droughts? I think that any paper by Anne Van Loon (e.g., Van Loon, 2015) may be useful to clarify this point.

*Thank you for the suggested references. In this sentence we are referring to seasonal precipitation levels lower than the historical norm, which may represent droughts. However, as this paragraph aims to overview climate teleconnections relevant to the Central Asian region, rather than droughts, we find it challenging to refer to Van Loon (2015) in this specific context.*

13. L88-89: This approach was proposed and tested more than two decades ago (e.g., Piechota et al., 1998).

In this context, the noted sentence refers to a general principle of combining initial hydrological conditions with future climate, rather than specifically to the earliest case where this was tested. We suggest keeping the reference unchanged. While Piechota et al. (1998) provides an interesting case study, it appears to rely primarily on predictors that characterize future climate.

14. L97: It would be good clarifying here that SWE can be directly obtained from reanalysis, or estimated by combining satellite remotely sensed snow depth and a snow density model.

15. L99-101: Please note that ensemble techniques have been used for decades in seasonal streamflow forecasting (e.g., Twedt et al., 1977; Day, 1985; Regonda et al., 2006; Wang et al., 2011; Arnal et al., 2018; Emerton et al., 2018; Lucatero et al., 2018; Girons Lopez et al., 2021; Araya et al., 2023).

Thank you for these suggestions. We expanded the literature review on use of ensemble approaches now as a separate dedicated paragraph (lines 78-92). Since there are numerous studies that use this approach, we focused primarily on those that resemble contextual similarities, specifically data-driven streamflow forecasting in snowmelt-dominated regions.

16. Table 1: I suggest adding the period used to compute the variables and more hydroclimatic descriptors, like mean annual runoff (mm/yr), mean annual runoff ratio and aridity index. Please change the units of seasonal discharge to mm/yr,

We appreciate this suggestion as it allowed us to detect some inconsistencies in the precipitation data. Due to absence of in-situ data, we estimated basin-averaged precipitation in Table 1 using the CHELSA-W5E5 dataset. However, when comparing mean annual runoff and precipitation estimates per mm/m2 using this data, in three basins (Varzob, Vaksh and Kashkadarya) the runoff coefficient surpasses 1 (Table R1). We assume that this discrepancy may be reasoned by underestimation of the CHELSA-W5E5 precipitation at least over those basins. To check this, we retrieved monthly station precipitation data from the recently published GHCN dataset (Applequist et al., 2024). It should be noted that available stations in the database do not cover all the basins, furthermore the stations monthly timeseries are largely fragmented in this domain (i.e. contain significant gaps). For the three mentioned basin the database contains only a station within Vaksh basin, with 17 to 19 complete observations per each month from 1979 to 2016 (temporal coverage of CHELSA-W5E5 dataset). Figure R1 below compares median monthly precipitation between station and corresponding CHELSA-W5E5 cell. While the annual cycle follows the same signature, the comparison suggests that CHELSA-W5E5 tends to underestimate precipitation across all months except August when precipitation is the lowest in this part of the domain. On average the annual bias of CHELSA-W5E5 estimate for this location constitutes around -25%. However, it is unclear whether station records are corrected for precipitation undercatch, which implies that the discrepancy could be even larger. In addition, this station is located at an elevation of 1,319 m.a.s.l., significantly below the basin's average elevation (3,530 m.a.s.l.), and we don't know the magnitude of bias at higher elevations in the catchment.

**Table R1.** Basin mean precipitation, runoff and runoff ratio estimated using CHELSA-W5E5 precipitation dataset

| Basin | Mean annual precipitation (mm) | Mean annual runoff (mm) | Runoff ratio |
|---|---|---|---|
| 1.Murghap | 319 | 31 | 0.10 |
| 2.Amudarya | 380 | 129 | 0.34 |
| 3.Varzob | 654 | 1121 | 1.71 |
| 4.Vaksh | 530 | 644 | 1.22 |
| 5.Kashkadarya | 530 | 1023 | 1.93 |
| 6.Zarafshan | 516 | 424 | 0.82 |
| 7.Naryn | 392 | 235 | 0.60 |
| 8.Chu | 391 | 186 | 0.48 |

[Figure]

**Figure R1**. Comparison of monthly precipitation between Station observations and corresponding CHELSA W5E5 data in the Vaksh Basin over 1979-2016. Station GHCN ID: TI000038851 (GARM/RASHT), latitude = 39.02, longitude = 70.36, elevation = 1316 m.a.s.l.

The revealed discrepancy does not affect the main findings of the study, as inter-annual peak season precipitation was used only to determine correlations with climate indices (Subsection "5.2 Association between climate oscillations and hydroclimatic variability across the study catchments"). A similar analysis using seasonal discharge shows almost identical patterns (Figure 3 in the original version of the manuscript) which suggests that the CHELSA-W5E5 dataset plausibly captures interannual variability in precipitation.

In this context, we believe it would be misleading to report basin-averaged precipitation in Table 1, and consequently, we cannot use these estimates to calculate the runoff ratio or other metrics such as the Aridity Index. Therefore, we have removed the respective column from Table 1. Additionally, since the CHELSA-W5E5 estimates are only available until 2016, we decided to use precipitation data from the TerraClimate dataset (Abatzoglou et al., 2018),) for displaying annual cycles across basins in Figure 1 and for estimating correlations in Subsection 5.2. The TerraClimate dataset covers the entire temporal range of discharge data across all basins, though it exhibits similar bias as CHELSA-W5E5 (not shown here). While this replacement does not affect the respective findings, as the annual cycles and correlations with climate indices show identical patterns, it is more appropriate for ensuring consistent temporal coverage.

> 17. L173: what link function did you use in your GLM?

We applied a Gaussian link function for the GLM model, now specified in the same line (59)

> 18. L189: Looks like the SVR works as a post-processor, right?

Yes, the SVR functions as a post-processor in our ensemble stacking approach.

> 19. L190-191: given the small sample size, I recommend deleting this step from your workflow (see comment #3).

Thank you for the suggestion. We revised our validation strategy in line with your recommendation in comment #3. All relevant sections of the manuscript were amended accordingly.

> 20. L205, L206, L297, L351 and L353 and everywhere else: the authors use the term "assimilate" when referring to the use of modeled SWE as a predictor in their statistical model. Nevertheless, such term is typically used when referring to a family of techniques that combine imperfect models with uncertain observations to improve dynamical model estimates (e.g., Liu and Gupta, 2007; Reichle, 2008; Kumar et al., 2016; Smyth et al., 2022). Since the authors do not refer to the former concept anywhere in this manuscript, I suggest deleting the words "assimilate" or "assimilation".

Thank you for highlighting this inconsistency in used terms. We amended the terms used accordingly throughout the text.

> 21. L221: what do you mean with the word "underperforming"?

> 22. L221-222: I think that this sentence contradicts the previous one. Also, if ERA5-L and MSWX are better, why don't you just pick one of these products for subsequent analyses? Some of your subsequent figures are unnecessarily complicated.

Thank you for this comment. We amended this sentence so that now reads as: "*While SWE estimates based on ERA5-L and MSWX generally show a higher correlation with seasonal discharge across most catchments, though in the absence of in-situ snow measurements, it is impossible to assert which of the four SWE estimates is relatively more consistent.*" (lines 307-309). Figure 3 is intended not only to illustrate the association between snowpack and seasonal streamflow and how this relationship changes across forecast issue dates, but also to highlight the differences between the snow estimates.

> 23. Section 5.2 and Figure 4: since your target variable is seasonal streamflow, you could show correlation results between this variable and climate indices here, and move the correlation results with precipitation to supplementary material.

We appreciate this suggestion. We propose retaining the correlation graph between peak-season precipitation and climate indices in the main text, while moving the streamflow correlation graph to the supplementary material. We believe this graph provides valuable context, which we also reflect on in the Summary and Discussion section in lines 532-534. We moved the other graph to Supplement (Figure S1)

> 24. Figure 5: I do not think you can support more than three predictors with a sample size n = 18 (see comment #3).

We now use up to three predictors per basin, with corresponding updates made across the text and figures.

> 25. L295: Do you mean winner among statistical models? Can you please be more specific?

Thank you for your comment. The sentence now reads "*There are no model types that are consistently superior in terms of performance across lead times, especially for the final (April 1st) forecast.*"

> 26. L301-302: I do not think that the authors are quantifying uncertainty (see comment #2).

> 27. Figure 6 is quite difficult to read. Since the focus of the paper is on the relevance of climate information in seasonal streamflow forecasting, why don't you just show the best-performing statistical model, with the best SWE product? Further, you should include the assessment period in each figure caption.

Thank you for your suggestion. We would like to retain Figure 6, as it not only displays the accuracy of both the base model forecasts and the final ensemble forecast, but it also illustrates the varying performance of the base models

across different issue dates. It also conveys the message that the ensemble forecast outperforms single model forecasts. We believe this broader comparison is useful for demonstrating the added value of the ensemble approach. Regarding the assessment periods, since they are now indicated in the Introduction and Data section, as well as in the caption to Figure 1., we do not see the need to repeat them again in this figure.

28. L313: This is not true for all catchments. See, for example, the red bars for the Kashkadarya and Chu basins.

With the revised version now incorporating all observations in the LOOCV and no hold-out validation sample, both the reported results and figures have been updated accordingly.

29. L322: Do you mean larger errors? Are you comparing against the results obtained with SWE and climate information? In that case, I really think you should define a Skill Score for a comparative assessment.

Thank you for the comment. Yes, we are comparing two configurations of the same models—one using only SWE and the other using both SWE and climate indices. To ensure a more consistent comparison, and in line with your previous suggestion (i.e. removing hold-out validation), there is now a single graph that compares normalized MAEs for both configurations for each basin and issue date.

30. Figure 8: I recommend presenting these results using scatter plots (eight panels), along with the 1:1 line, percent bias, MAE and $R^2$.

We appreciate this suggestion. We displayed these results as Q-Q plots. We did not embed the three metrics, as they would obscure the plots, given that each basin plot includes up to four forecast issue dates, resulting in up to 12 text elements per plot. However, upon your request, we could present these results as a table in the Supplement.

31. L348: What do you mean with 'effectively'? That near real-time SWE estimates are actually useful for seasonal streamflow forecasting?

By 'effectively,' we intended to convey that SWE estimates derived from or modelled using global sources, despite their biases and spatio-temporal inconsistencies, can still provide added value for seasonal streamflow forecasting. While this point may seem trivial, we believe it is relevant in the context of forecasting without in-situ data on predictors. We revised this sentence to ensure clarity (line 514 in the tracked changes version).

32. L350: I do not think the authors have presented any uncertainty or error propagation analysis (please see comment #2)

We assume that the message of the sentence is now valid since it now also relies on incorporated uncertainty analysis.

33. L352: Did you actually assess the accuracy of SWE products using in-situ observations?

No, as we note in the Introduction (lines_137-41) systematic in-situ SWE measurements are absent in the region.

34. L420: In my opinion, models adjusted with such a small sample cannot be regarded as "reliable".

We changed the wording to 'plausible'.

***Suggested edits***

35. L28: "where it sustains" -> "sustaining".

36. L32: "Accurate water availability forecasts" -> "accurate water supply forecasts".

37. L36: "current hydrologic conditions" -> "initial hydrologic conditions".

38. L42: "multiple variables" -> "multiple predictor variables".

39. L43: delete "the context of".

40. L61 and L63: replace "from now on" by "hereafter".

41. L67: delete "from satellite".

42. L73: "ENSO in its cold phase" -> "the cold phase of ENSO".

43. L74: delete "ENSO's".

44. L95-96: "used to conduct" -> "conducted".

45. L124: I think that the right word is "predictand".

46. L130: delete "in near real-time".

47. L132: "we simulated" -> "we obtained".

48. L155-156: "precipitation levels" -> "precipitation amounts".

49. L174: add "SVR" after "support vector regression".

50. L214-215: I suggest deleting this sentence.

Thank you for the proposed edits. We updated these parts in line with your suggestions.

References:

Abatzoglou, J. T., Dobrowski, S. Z., Parks, S. A., and Hegewisch, K. C.: TerraClimate, a high-resolution global dataset of monthly climate and climatic water balance from 1958–2015, Sci. Data, 5, 170191, https://doi.org/10.1038/sdata.2017.191, 2018.

Apel, H., Abdykerimova, Z., Agalhanova, M., Baimaganbetov, A., Gavrilenko, N., Gerlitz, L., Kalashnikova, O., Unger-Shayesteh, K., Vorogushyn, S., and Gafurov, A.: Statistical forecast of seasonal discharge in Central Asia using observational records: development of a generic linear modelling tool for operational water resource management, Hydrol. Earth Syst. Sci., 22, 2225–2254, https://doi.org/10.5194/hess-22-2225-2018, 2018.

Applequist, S., Durre, I., and Vose, R.: The Global Historical Climatology Network Monthly Precipitation Dataset, Version 4, Sci. Data, 11, 633, https://doi.org/10.1038/s41597-024-03457-z, 2024.

Dietterich, T. G.: Ensemble Methods in Machine Learning, in: Multiple Classifier Systems, 1–15, 2000.

Immerzeel, W. W., Lutz, A. F., Andrade, M., Bahl, A., Biemans, H., Bolch, T., Hyde, S., Brumby, S., Davies, B. J., Elmore, A. C., Emmer, A., Feng, M., Fernández, A., Haritashya, U., Kargel, J. S., Koppes, M., Kraaijenbrink, P. D. A., Kulkarni, A. V., Mayewski, P. A., Nepal, S., Pacheco, P., Painter, T. H., Pellicciotti, F., Rajaram, H., Rupper, S., Sinisalo, A., Shrestha, A. B., Viviroli, D., Wada, Y., Xiao, C., Yao, T., and Baillie, J. E. M.: Importance and vulnerability of the world's water towers, Nature, 577, 364–369, https://doi.org/10.1038/s41586-019-1822-y, 2020.

Kiers, H. A. L. and Smilde, A. K.: A comparison of various methods for multivariate regression with highly collinear variables, Stat. Methods Appl., 16, 193–228, https://doi.org/10.1007/s10260-006-0025-5, 2007.

Van Loon, A. F.: Hydrological drought explained, Wiley Interdiscip. Rev. Water, 2, 359–392, https://doi.org/10.1002/wat2.1085, 2015.

Mendoza, P. A., Wood, A. W., Clark, E., Rothwell, E., Clark, M. P., Nijssen, B., Brekke, L. D., and Arnold, J. R.: An intercomparison of approaches for improving operational seasonal streamflow forecasts, Hydrol. Earth Syst. Sci., 21, 3915–3935, https://doi.org/10.5194/hess-21-3915-2017, 2017.

Zounemat-Kermani, M., Batelaan, O., Fadaee, M., and Hinkelmann, R.: Ensemble machine learning paradigms in hydrology: A review, J. Hydrol., 598, 126266, https://doi.org/https://doi.org/10.1016/j.jhydrol.2021.126266, 2021.

---

## Author Response (AR2)

**Dear Reviewers,**

Thank you for your valuable feedback. Please find below your referee comments (in black) and our responses (in blue).

**With regards, the authors.**

**RC: Reviewer #1**

Review of "The Value of Hydroclimatic Teleconnections for Snow-based Seasonal Streamflow Forecasting in Central Asia" (revised manuscript) by Umirbekov et al.:

This applied research article synthesizes and integrates a variety of methods and data types to create a convincing set of seasonal water supply forecast models for several rivers in Central Asia. The work is innovative, well-thought-out, technically sound, and socially relevant given its potential to support water management in an under-served region that would benefit from this kind of operational forecast system. Free public availability of the author's code and all the required input data means the forecast model could be quickly and easily implemented in production systems. The paper additionally presents a number of original research outcomes that should inform other studies around interactions between snowpack data, atmosphere-ocean circulation patterns, water supply forecasting, and practical machine learning systems.

In my opinion, this paper will be a strong contribution to HESS, and I recommend publication as-is (or with some very minor additional revisions). The revised article fully addresses the review comments I provided on the original submission, and I have only a few small additional suggestions to make here:

**We appreciate your overall feedback on the revised manuscript.**

• First sentence of the abstract: the phrasing is awkward – I suggest improving this to read something more like, "Due to the long memory of snow processes, statistically based seasonal streamflow prediction models in snow-dominated catchments can successfully leverage, but also typically rely on, snowpack estimates."

**We have amended the sentence accordingly.**

• Lines 36-44: This is much-improved over the original manuscript but would still benefit from a little additional work. A few points the authors might bear in mind: both process-simulation and data-driven water supply forecast models can ingest seasonal to subseasonal climate forecasts as input (see for example Lehner et al., Geophysical Research Letters, 2017, https://doi.org/10.1002/2017GL076043); process-based models have the advantage of explicitly capturing process physics, enhancing credibility and interpretability; data-driven models have the advantage of not requiring assumptions about the relevant physics and how to represent it in a pragmatic computational model. Rewriting the passage just a little more to acknowledge these points would lend greater credibility to the article overall.

**We have amended the paragraph with additional excerpt along those lines (lines 44-48 in the track-changes version of the manuscript)**

• Lines 176-186: it might be worth explicitly mentioning here that while several models were used in the ensemble, each of them individually is quite simple and parsimonious, with a single target variable (seasonal discharge volume) and three or fewer input variables (as subsequently noted on lines 215-216). That means there's only a small number of parameters to estimate from the limited sample size. In other words, the methodology used here is suitable for application to short datasets. There is some precedent in water supply forecast modeling for deliberately fine-tuning statistical and machine learning architectures to maximize parsimony and minimize the number of parameters to be estimated, enabling application to short

datasets with good out-of-sample performance, as well as improved regularization and geophysical explainability (some examples are Fleming et al., 2021, 2024, which are already cited in the manuscript).

Thank you for this suggestion. We have amended the Method section with additional note (lines 261-263 in the track-changes version of the manuscript).

• I really like Figure 5, but I think there's a typographical error. The caption reads, "For example, the April 1st forecast models for the Amudarya use as predictors the SWE estimate as of the beginning of March, the state of the PDO index in November, and the SCAN index in February." I think the figure states, though, that these models use the SCAN index in January, not February, for that forecast date and river.

Thank you for pointing at this mistake. We have corrected the wording in the caption.

**Reviewer #2**

I want to thank the authors for incorporating most of the suggestions provided in the first round of revisions. Despite the manuscript has been improved considerably, I have a suite of comments that I would like the authors to address before this paper is accepted for publication.

We appreciate your feedback on our previous changes to the manuscript.

Minor comments

1. L41: I think it would be more appropriate re-writing as "...some process-based models have higher computational demand (e.g., Oleson et al., 2010; Niu et al., 2011; Clark et al., 2021)...".

Thank you for this suggestion. We believe that the current version of the sentence is justified, as the context compares the computational demand of process-based and data-driven models. The following references may serve as supporting evidence (Clark et al., 2017; Tsai et al., 2021; Chen et al., 2022; Rahman et al., 2022; Bennett et al., 2024).

2. L88, L312 and everywhere else: please define the winter season (DJF?), since most readers will not be familiar with your study domain. The same comment applies to the remaining seasons.

This sentence is part of a general introductory paragraph on common hydrological cycle pattern in Central Asia. We define all relevant seasons we focus on as a range of respective months in the following paragraph (lines 99-101), such as "*cold season*" and "*growing season*", and use these terms throughout the text. In our opinion, specifying months for the terms "*winter*", "*spring*", and "*summer*" in the preceding sentence may introduce confusion for readers, as they do not fully align with the "*cold*" and "*growing*" seasons that are the focus of our study.

3. L131, L230 and everywhere else: I advise replacing "prediction" with "hindcast", since you are actually presenting results from retrospective forecasting (i.e., hindcasting) experiments. Please be precise and consistent with the terminology.

Thank you for this suggestion. The sentence in question provides a general explanation of suggested approach for seasonal streamflow forecasting in the region. In our opinion, using "*hindcast*" in this context would be confusing. For the same reason, we continue to use the term "*prediction*" in the Methods section, which details the seasonal streamflow model based on ensemble stacking. However, we acknowledge that we incorrectly referred to results as "*forecasts*" and have ensured that in the revised version of the manuscript this has been corrected to "*hindcasts*" where appropriate.

4. L233-234: if I understood well, you assess – for each model – cross-validated (deterministic?) hindcasts to get 16 evaluation metrics, and then select the k < 16 that fulfill the requirement R2>0.2, right? Please clarify.

Yes, that is correct. For each of the 16 base models, we compute a leave-one-out cross-validated (LOOCV)  $R^2$  value based on deterministic hindcasts. Only base models that achieve an LOOCV  $R^2 > 0.2$  are retained for further analysis. We have amended these lines (240-243) in the revised text for a better clarity.

5. Section 4.1: Did you try correlating seasonal averages (i.e., temporally averaged 2-month, 3-month, etc.) of your climate indices against seasonal precipitation for predictor screening?

Thank you for guiding question. No, we did not apply temporal averaging for predictor screening. However, we agree that averaging may be more appropriate for oscillations with relatively slower dynamics, such as SOI and PDO. Implementing this approach would require modifications to our code and could alter predictor lags, potentially affecting model structure and comparability. To maintain consistency and avoid introducing additional complexity at this stage, we prefer to keep the current methodology unchanged.

7. I strongly advise the authors to move Figure S1 to the main manuscript and move Figure 4 to the supplement, since your paper is about seasonal streamflow (and NOT precipitation) forecasting. Also, the current Figures S1 and 4 are nearly identical.

Thank you for this suggestion. However, we think that Figure 4 should remain in the main manuscript, as it is directly tied to our conceptual framework. Our approach uses climate oscillations as approximators for precipitation variability in the upcoming season. We believe that emphasizing the climate oscillation–precipitation–streamflow link provides a clearer justification for incorporating climate oscillations. Figure S1, on the other hand, may be viewed as supporting evidence that climate oscillations influence streamflow by modulating precipitation. We acknowledge that the initial version of the manuscript contained a paragraph explicitly explaining this rationale, which was removed during a previous revision. To address this, we have now revised the Introduction in lines 130-134 to better clarify our conceptual framework.

8. L339-342: the statements concerning the numbers of models are very hard to visualize. I recommend adding those numbers in Figure 6 for each forecast initialization.

We appreciate this suggestion. Since Figure 6 is already stacked, adding additional numbers directly to the figure may reduce readability. Instead, we propose moving information on the number and types of models to the Supplementary Material, ensuring that it is appropriately referenced in the text for clarity (lines 364-365).

9. L351: I presume you are referring to R2 results here, right? I do not think you should refer to "uncertainty", since you are not quantifying forecast spread or providing confidence intervals. Please be more precise and refer to what you are actually showing.

Yes, this comment refers to the evaluation of models in terms of  $R^2$  and nMAE. We have revised the sentence to use the term "*lower accuracy*" for better clarity. Thank you for this correction.

10. Figure 7: Please clarify whether you are showing the results from the meta-learner model.

Yes, these results are from the meta-learner models. We have revised the caption accordingly.

11. Figure 8: Are you displaying the ensemble hindcast mean along the y-axis? Please clarify. Also, these results should not be in sub-section 5.4, since you are not illustrating any predictive uncertainty.

Figure 8 displays hindcasts produced by the meta-learner model. These results are now moved to sub-section 5.3

12. L398-399: This description should be in the methods section. How many times did you resample the data? Note that this step would be redundant if the SVM meta learner produced ensemble hindcasts.

We have moved the relevant description to the Methods section as a new sub-section titled "Uncertainty Estimation" (lines 265-268). In the earlier approach, we bootstrapped only the base model predictions, which primarily captured uncertainty stemming from the ensemble structure. We have now implemented a different method: after resampling the input data, we retrain both the base models and the meta-learner for each of the 500 bootstrap iterations.

13. L401: I strongly advise the authors to be more quantitative when judging the "width of uncertainty bounds". To this end, they can compute the alpha reliability index (Renard et al., 2010), as in previous seasonal hindcasting studies (e.g., Mendoza et al., 2017; Araya et al., 2023).

Thank you for this suggestion. We have now amended the wording in that and other lines. We have also amended this section with the following sentence: "It should be noted that the uncertainty intervals are estimated by bootstrapping a relatively short streamflow time series and do not account for uncertainty caused by the potential limited representativeness of the actual natural variability by the observations."

14. L409: please clarify what you mean with "more consistent".

We have now replaced "more consistent" with "relatively less uncertain".

15. L139, L391, L406 and everywhere else: please avoid using "significant" or "significance", unless you refer to statistically significant result.

We have now replaced "significant" in those lines with other words.

16. L458-459: "The resulting forecast models generate credible simulations...". Your models are producing hindcasts and NOT simulations (please revise Beven and Young, 2013). Also, the sentence reads as overselling, since your results are not good for all lead times. I suggest deleting.

We now replaced "simulations" with "hindcasts". Please note that the sentence notes "albeit with performance variations depending on lead time.". We do not consider the message as overselling, given the results we present and multiple restrictions we encounter within this study.

17. L464-465: "In most catchments, the SOI, PDO, or both were utilised, indicating the dominant influence of ENSO". The Pacific Decadal Oscillation (PDO) can modulate ENSO, but PDO and ENSO are different modes of variability. I suggest re-wording to avoid confusing readers.

The sentence is amended by adding: "..and other climate variability patterns in the Pacific Ocean."

18. L490: I think what you should write here is "more accurate seasonal streamflow hindcasts". Note that the term "reliable" has a very specific connotation in probabilistic forecasting, and is related with the degree to which forecast probabilities match relative observed frequencies (see Wilks, 2019).

**We have replaced "reliable" with "accurate".**

Suggested edits

- 19. L28: delete "other".
- 20. L33: Add "Additionally," before "accurate".

21. L42: revise "data-drivenapproaches".

22. L44 and L47: I suggest deleting "primarily".

23. L63: "multi ensemble" -> "multi-model ensemble".

24. L75: higher prediction accuracy and better quantify -> "the quantification of".

25. L94-95: "One approach" -> "The first approach".

26. L111-112: "for forecasting" -> "to forecast".

27. L331: I suggest rewriting as "...with varying R2...", since this is what you are actually showing.

Thank you for these suggestions. We have edited the text accordingly in most of the suggested instances, except the comment #20.

**References:**

Bennett, A. *et al.* (2024) 'Spatio-Temporal Machine Learning for Regional to Continental Scale Terrestrial Hydrology', *Journal of Advances in Modeling Earth Systems*, 16(6), p. e2023MS004095. doi: https://doi.org/10.1029/2023MS004095.

Chen, X. *et al.* (2022) 'Comparison of deep learning models and a typical process-based model in glacio-hydrology simulation', *Journal of Hydrology*, 615, p. 128562. doi: https://doi.org/10.1016/j.jhydrol.2022.128562.

Clark, M. P. *et al.* (2017) 'The evolution of process-based hydrologic models: historical challenges and the collective quest for physical realism', *Hydrology and Earth System Sciences*, 21(7), pp. 3427–3440. doi: 10.5194/hess-21-3427-2017.

Rahman, K. U. *et al.* (2022) 'Comparison of machine learning and process-based SWAT model in simulating streamflow in the Upper Indus Basin', *Applied Water Science*, 12(8), p. 178. doi: 10.1007/s13201-022-01692-6.

Tsai, W.-P. *et al.* (2021) 'From calibration to parameter learning: Harnessing the scaling effects of big data in geoscientific modeling', *Nature Communications*, 12(1), p. 5988. doi: 10.1038/s41467-021-26107-z.